# LEARNING TO EMBED TIME SERIES PATCHES INDEPENDENTLY

**Seunghan Lee, Taeyoung Park, Kibok Lee**
Department of Statistics and Data Science, Yonsei University
{seunghan9613,tpark,kibok}@yonsei.ac.kr

## ABSTRACT

Masked time series modeling has recently gained much attention as a self-supervised representation learning strategy for time series. Inspired by masked image modeling in computer vision, recent works first patchify and partially mask out time series, and then train Transformers to capture the dependencies between patches by predicting masked patches from unmasked patches. However, we argue that capturing such patch dependencies might not be an optimal strategy for time series representation learning; rather, learning to embed patches independently results in better time series representations. Specifically, we propose to use 1) the simple patch reconstruction task, which autoencode each patch without looking at other patches, and 2) the simple patch-wise MLP that embeds each patch independently. In addition, we introduce complementary contrastive learning to hierarchically capture adjacent time series information efficiently. Our proposed method improves time series forecasting and classification performance compared to state-of-the-art Transformer-based models, while it is more efficient in terms of the number of parameters and training/inference time. Code is available at this repository: https://github.com/seunghan96/pits.

## 1 INTRODUCTION

Time series (TS) data finds application in a range of downstream tasks, including forecasting, classification, and anomaly detection. Deep learning has shown its superior performance in TS analysis, where learning good representations is crucial to the success of deep learning, and self-supervised learning has emerged as a promising strategy for harnessing unlabeled data effectively. Notably, contrastive learning (CL) and masked modeling (MM) have demonstrated impressive performance in TS analysis as well as other domains such as natural language processing (Devlin et al., 2018; Brown et al., 2020) and computer vision (Chen et al., 2020; Dosovitskiy et al., 2021).

Masked time series modeling (MTM) task partially masks out TS and predicts the masked parts from the unmasked parts using encoders capturing dependencies among the patches, such as Transformers (Zerveas et al., 2021; Nie et al., 2023). However, we argue that learning such dependencies among patches, e.g., predicting the unmasked parts based on the masked parts and utilizing architectures capturing dependencies among the patches, might not be necessary for representation learning.

To this end, we introduce the concept of *patch independence* which does not consider the interaction between TS patches when embedding them. This concept is realized through two key aspects: 1) the pretraining task and 2) the model architecture. Firstly, we propose a patch reconstruction task that reconstructs the unmasked patches, unlike the conventional MM that predicts the masked ones. We refer to these tasks as the patch-independent (PI) task and the patch-dependent (PD) task, respectively, as the former does not require information about other patches to reconstruct each patch, while

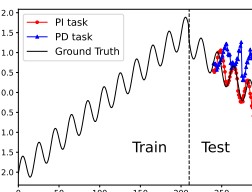

Figure 1: PI vs. PD.

the latter does. Figure 1 illustrates a toy example of TS forecasting. While the Transformer pretrained on the PD task (Nie et al., 2023) fails to predict test data under distribution shift, the one pretrained on the PI task is robust to it. Secondly, we employ the simple PI architecture (e.g., MLP), exhibiting better efficiency and performance than the conventional PD architecture (e.g., Transformer).

In this paper, we propose **P**atch **I**ndependence for **T**ime **S**eries (*PITS*), which utilizes unmasked patch reconstruction as the PI pretraining task and MLP as the PI architecture. On top of that, we introduce complementary CL to efficiently capture adjacent time series information, where CL is performed using two augmented views of original samples that are masked in complementary ways.

| | | CL for TS* | TST (KDD 2021) | TS2Vec (AAAI 2022) | FEDFormer (ICML, 2022) | DLinear (AAAI 2023) | PatchTST (ICLR 2023) | TimeMAE (arXiv 2023) | SimMTM (NeurIPS 2023) | PITS (Ours) |
|---|---|---|---|---|---|---|---|---|---|---|
| Pretraining method | CL | ✓ | | ✓ | | | | | | ✓ |
| | MTM | | ✓ | | | | ✓ | ✓ | ✓ | ✓ |
| | No (Sup.) | | ✓ | | ✓ | ✓ | ✓ | | ✓ | ✓ |
| Downstream task | Forecasting | | ✓ | ✓ | ✓ | ✓ | ✓ | | ✓ | ✓ |
| | Classification | ✓ | ✓ | ✓ | | | | ✓ | ✓ | ✓ |

\* T-Loss (NeurIPS 2019), Self-Time (arXiv 2020), TS-SD (IJCNN 2021), TS-TCC (IJCAI 2021), TNC (ICLR 2021), Mixing-up (PR Letters 2022), TF-C (NeurIPS 2022), TimeCLR (KBS 2022), CA-TCC (TPAMI 2023).

Table 1: Comparison table of SOTA methods in TS.

We conduct extensive experiments on various tasks, demonstrating that our proposed method outperforms the state-of-the-art (SOTA) performance in both forecasting and classification tasks, under both standard and transfer learning settings. The main contributions are summarized as follows:

- We argue that *learning to embed time series patches independently* is superior to learning them dependently for TS representation learning, in terms of both performance and efficiency. To achieve patch independence, we propose PITS, which incorporates two major modifications on the MTM: 1) to make the task patch-independent, reconstructing the unmasked patches instead of predicting the masked ones, and 2) to make the encoder patch-independent, eliminating the attention mechanism while retaining MLP to ignore correlation between the patches during encoding.

- We introduce complementary contrastive learning to hierarchically capture adjacent TS information efficiently, where positive pairs are made by complementary random masking.

- We present extensive experiments for both low-level forecasting and high-level classification, demonstrating that our method improves SOTA performance on various downstream tasks. Also, we discover that PI tasks outperforms PD tasks in managing distribution shifts, and that PI architecture is more interpretable and robust to patch size compared to PD architecture.

## 2 RELATED WORKS

**Self-supervised learning.** In recent years, self-supervised learning (SSL) has gained attention for learning powerful representations from unlabeled data across various domains. The success of SSL comes from active research on pretext tasks that predict a certain aspect of data without supervision. Next token prediction (Brown et al., 2020) and masked token prediction (Devlin et al., 2018) are commonly used in natural language processing, and jigsaw puzzles (Noroozi & Favaro, 2016) and rotation prediction (Gidaris & Komodakis, 2018) are commonly used in computer vision.

Recently, contrastive learning (CL) (Hadsell et al., 2006) has emerged as an effective pretext task. The key principle of CL is to maximize similarities between positive pairs while minimizing similarities between negative pairs (Gao et al., 2021; Chen et al., 2020; Yue et al., 2022). Another promising technique is masked modeling (MM), which trains the models to reconstruct masked patches based on the unmasked part. For instance, in natural language processing, models predict masked words within a sentence (Devlin et al., 2018), while in computer vision, they predict masked patches in images (Baevski et al., 2022; He et al., 2022; Xie et al., 2022) within their respective domains.

**Masked time series modeling.** Besides CL, MM has gained attention as a pretext task for SSL in TS. This task involves masking a portion of the TS and predicting the missing values, known as masked time series modeling (MTM). While CL has shown impressive performance in high-level classification tasks, MM has excelled in low-level forecasting tasks (Yue et al., 2022; Nie et al., 2023). TST (Zerveas et al., 2021) applies the MM paradigm to TS, aiming to reconstruct masked timestamps. PatchTST (Nie et al., 2023) focuses on predicting masked subseries-level patches to capture local semantic information and reduce memory usage. SimMTM (Dong et al., 2023) reconstructs the original TS from multiple masked TS. TimeMAE (Cheng et al., 2023) trains a transformer-based encoder using two pretext tasks, masked codeword classification and masked representation regression. Table 1 compares various methods in TS including ours in terms of two criterions: pretraining methods and downstream tasks, where *No (Sup.)* in Pretraining method indicates a supervised learning method that does not employ pretraining.

Different from recent MTM works, we propose to reconstruct unmasked patches through autoencoding. A primary concern on autoencoding is the trivial solution of identity mapping, such that the dimension of hidden layers should be smaller than the input. To alleviate this, we introduce dropout after intermediate fully-connected (FC) layers, which is similar to the case of stacked denoising autoencoders (Liang & Liu, 2015), where the ablation study can be found in Figure 4.

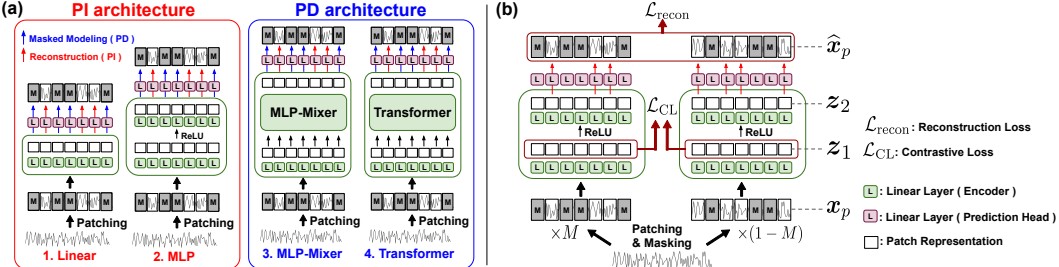

Figure 2: **Patch-independent strategy of PITS.** (a) illustrates the pretraining tasks and encoder architectures in terms of PI and PD. (b) demonstrates the proposed PITS, which utilizes a PI task with a PI architecture. TS is divided into patches and augmented with complementary masking. Representations from the 1st and 2nd layers of MLP is used for CL and the reconstruction, respectively.

**Combination of CL and MM.** There have been recent efforts to combine CL and MM for representation learning (Jiang et al., 2023; Yi et al., 2023; Huang et al., 2022; Gong et al., 2023; Dong et al., 2023). Among these works, SimMTM (Dong et al., 2023) addresses an MM task with a regularizer in its objective function in the form of a contrastive loss. However, it differs from our work in that it focuses on CL between TS, while our proposed CL operates with patches within a single TS.

**Complementary masking.** SdAE (Chen et al., 2022) employs a student branch for information reconstruction and a teacher branch to generate latent representations of masked tokens, utilizing a complementary multi-fold masking strategy to maintain relevant mutual information between the branches. TSCAE (Ye et al., 2023) addresses the gap between upstream and downstream mismatches in the pretraining model based on MM by introducing complementary masks for teacher-student networks, and CFM (Liao et al., 2022) introduces a trainable complementary masking strategy for feature selection. Our proposed complementary masking strategy differs in that it is not designed for a distillation model, and our masks are not learnable but randomly generated.

**Linear models for time series forecasting.** Transformer (Vaswani et al., 2017) is a popular sequence modeling architecture that has prompted a surge in Transformer-based solutions for time series analysis (Wen et al., 2022). Transformers derive their primary strength from the multi-head self-attention mechanism, excelling at extracting semantic correlations within extensive sequences. Nevertheless, recent work by Zeng et al. (2023) shows that simple linear models can still extract such information captured by Transformer-based methods. Motivated by this work, we propose to use a simple MLP architecture that does not encode interaction between time series patches.

## 3 METHODS

We address the task of learning an embedding function $f_\theta : \boldsymbol{x}_p^{(i,c,n)} \to \boldsymbol{z}^{(i,c,n)}$ for a TS patch where $\boldsymbol{x}_p = \left\{ \boldsymbol{x}_p^{(i,c,n)} \right\}$, $\boldsymbol{z} = \left\{ \boldsymbol{z}^{(i,c,n)} \right\}$, and $i = 1, \ldots, B$, $c = 1, \ldots, C$, $n = 1, \ldots, N$. Here, $B$, $C$, $N$ are the number of TS, number of channels in a single TS, and number of patches in a single channel of a single TS. The input and the output dimension, which are the patch size and patch embedding dimension, are denoted as $P$ and $D$, respectively, i.e., $\boldsymbol{x}_p^{(i,c,n)} \in \mathbb{R}^P$ and $\boldsymbol{z}^{(i,c,n)} \in \mathbb{R}^D$. Our goal is to learn $f_\theta$ extracting representations that perform well on various downstream tasks.

**Channel independence & Patch independence.** We use the channel independence architecture for our method, where all channels share the same model weights and embedded independently, i.e, $f_\theta$ is independent to $c$. This has shown robust prediction to the distribution shift compared to channel-dependent approaches (Han et al., 2023). Also, we propose to use the PI architecture, where all patches share the same model weights and embedded independently, i.e, $f_\theta$ is independent to $n$. We illustrate four different PI/PD architectures in Figure 2(a), where we use MLP for our proposed PITS, due to its efficiency and performance, as demonstrated in Table 13 and Table 7, respectively.

### 3.1 PATCH-INDEPENDENT TASK: PATCH RECONSTRUCTION

Unlike the conventional MM task (i.e., PD task) that predicts masked patches using unmasked ones, we propose the patch reconstruction task (i.e., PI task) that autoencodes each patch without looking at the other patches, as depicted in Figure 2(a). Hence, while the original PD task requires capturing patch dependencies, our proposed task does not. A patchified univariate TS can be reconstructed in

two different ways[1]: 1) reconstruction at once by a FC layer processing the concatenation of patch representations: $\text{concat}\left(\widehat{\boldsymbol{x}}_p^{(i,c,:)}\right) = W_1 \text{concat}\left(\boldsymbol{z}^{(i,c,:)}\right)$ where $W_1 \in \mathbb{R}^{N \cdot P \times N \cdot D}$, and 2) patch-wise reconstruction by a FC layer processing each patch representation: $\widehat{\boldsymbol{x}}_p^{(i,c,n)} = W\boldsymbol{z}^{(i,c,n)}$ where $W \in \mathbb{R}^{P \times D}$. Similar to Nie et al. (2023), we employ the patch-wise reconstruction which yields better performance across experiments.

## 3.2 PATCH-INDEPENDENT ARCHITECTURE: MLP

While MTM has been usually studied with Transformers for capturing dependencies between patches, we argue that learning to embed patches independently is better. Following this idea, we propose to use the simple PI architecture, so that the encoder solely focuses on extracting patch-wise representations. Figure 2(a) shows the examples of PI/PD pretraining tasks and encoder architectures. For PI architectures, **Linear** consists of a single FC layer model and **MLP** consists of a two-layer MLP with ReLU. For PD architectures, **MLP-Mixer**[2] (Tolstikhin et al., 2021; Chen et al., 2023) consists of a single FC layer for time-mixing ($N$-dim) followed by a two-layer MLP for patch-mixing ($D$-dim), and **Transformer** consists of a self-attention layer followed by a two-layer MLP, following Nie et al. (2023). The comparison of the efficiency between MLP and Transformer in terms of the number of parameters and training/inference time is provided in Table 13.

## 3.3 COMPLEMENTARY CONTRASTIVE LEARNING

To further boost performance of learned representations, we propose complementary CL to hierarchically capture adjacent TS information. CL requires two views to generate positive pairs, and we achieve this by a complementary masking strategy: for a TS $\boldsymbol{x}$ and a mask $\boldsymbol{m}$ with the same length, we consider $\boldsymbol{m} \odot \boldsymbol{x}$ and $(1 - \boldsymbol{m}) \odot \boldsymbol{x}$ as two views, where $\odot$ is the element-wise multiplication and we use 50% masking ratio for experiments. Note that the purpose of masking is to generate two views for CL; it does not affect the proposed PI task, and it does not require an additional forward pass when using the proposed PI architectures, such that the additional computational cost is negligible.

Figure 3 illustrates an example of complementary CL, where we perform CL hierarchically (Yue et al., 2022) by max-pooling on the patch representations along the temporal axis, and compute and aggregate losses computed at each level. Then, the model learns to find missing patch information in one view, by contrasting the similarity with another view and the others, so that the model can capture adjacent TS information hierarchically.

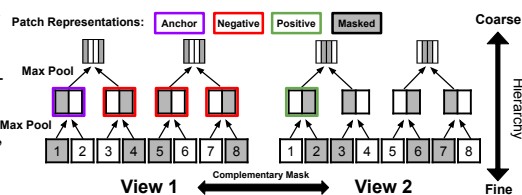

Figure 3: Complementary contrastive learning.

## 3.4 OBJECTIVE FUNCTION

As illustrated in Figure 2(b), we perform CL at the first layer and reconstruction by an additional projection head on top of the second layer, based on the ablation study in Table 9. To distinguish them, we denote representations obtained from the two layers in MLP as $\boldsymbol{z}_1$ and $\boldsymbol{z}_2$, respectively.

**Reconstruction loss.** As discussed in Section 3.1, we feed $\boldsymbol{z}_2$ into the patch-wise linear projection head to get a reconstructed result: $\widehat{\boldsymbol{x}}_p = W\boldsymbol{z}_2$. Then, the reconstruction loss can be written as:

$$\mathcal{L}_{\text{recon}} = \sum_{i=1}^{B}\sum_{c=1}^{C}\sum_{n=1}^{N} \left\| \boldsymbol{m}^{(i,c,n)} \odot \left( \boldsymbol{x}_p^{(i,c,n)} - \widehat{\boldsymbol{x}}_p^{(i,c,n)} \right) \right\|_2^2 + \left\| (1 - \boldsymbol{m}^{(i,c,n)}) \odot \left( \boldsymbol{x}_p^{(i,c,n)} - \widehat{\boldsymbol{x}}_p^{(i,c,n)} \right) \right\|_2^2$$

$$= \sum_{i=1}^{B}\sum_{i=1}^{C}\sum_{n=1}^{N} \left\| \boldsymbol{x}_p^{(i,c,n)} - \widehat{\boldsymbol{x}}_p^{(i,c,n)} \right\|_2^2, \tag{1}$$

where $\boldsymbol{m}^{(i,c,n)} = 0$ if the first view $\boldsymbol{x}_p^{(i,c,n)}$ is masked, and 1 otherwise. As derived in Eq. 1, the reconstruction task is not affected by complementary masking, i.e., reconstructing the unmasked patches of the two views is the same as reconstructing patches without complementary masking.

---

[1]Biases are omitted for conciseness.

[2]While TSMixer is a variation of MLP-Mixer proposed for TS concurrent to our work, we found that TSMixer does not perform well with SSL, so we use our own variation of MLP-Mixer here.

**Contrastive loss.** Inspired by the cross-entropy loss-like formulation of the contrastive loss in Lee et al. (2021), we establish a softmax probability for the relative similarity among all the similarities considered when computing temporal contrastive loss. For conciseness, let $z_1^{(i,c,n)} = z_1^{(i,c,n+2N)}$ and $z_1^{(i,c,n+N)}$ be the two views of the patch embedding $x^{(i,c,n)}$. Then, the softmax probability for a pair of patch indices $(n, n')$ is defined as:

$$p(i, c, (n, n')) = \frac{\exp(z_1^{(i,c,n)} \circ z_1^{(i,c,n')})}{\sum_{s=1,s \neq n}^{2N} \exp(z_1^{(i,c,n)} \circ z_1^{(i,c,s)})}, \tag{2}$$

where we use the dot product as the similarity measure $\circ$. Then, the total contrastive loss can be written as:

$$\mathcal{L}_{\mathrm{CL}} = \frac{1}{2BCN} \sum_{i=1}^{B} \sum_{i=1}^{C} \sum_{n=1}^{2N} -\log p(i, c, (n, n+N)), \tag{3}$$

where we compute the hierarchical losses by max-pooling $z^{(i,c,n)}$'s along with the dimension $n$ repeatedly with the following substitutions until $N = 1$:

$$z^{(i,c,n)} \leftarrow \mathrm{MaxPool}([z^{(i,c,2n-1)}, z^{(i,c,2n)}]), \quad N \leftarrow \lfloor N/2 \rfloor. \tag{4}$$

The final loss of PITS is the sum of the reconstruction loss and hierarchical contrastive loss:

$$\mathcal{L} = \mathcal{L}_{\mathrm{recon}} + \mathcal{L}_{\mathrm{CL}}. \tag{5}$$

**Instance normalization.** To mitigate the problem of distribution shift between the training and testing data, we normalize each TS with zero mean and unit standard deviation (Kim et al., 2021). Specifically, we normalize each TS before patching and mean and deviation are added back to the predicted output.

## 4 EXPERIMENTS

### 4.1 EXPERIMENTAL SETTINGS

**Tasks and evaluation metrics.** We demonstrate the effectiveness of the proposed PITS on two downstream tasks: time series forecasting (TSF) and classification (TSC) tasks. For evaluation, we mainly follow the standard SSL framework that pretrains and fine-tunes the model on the same dataset, but we also consider in-domain and cross-domain transfer learning settings in some experiments. As evaluation metrics, we use the mean squared error (MSE) and mean absolute error (MAE) for TSF, and accuracy, precision, recall, and the $F_1$ score for TSC.

### 4.2 TIME SERIES FORECASTING

**Datasets and baseline methods.** For forecasting tasks, we experiment seven datasets, including four ETT datasets (ETTh1, ETTh2, ETTm1, ETTm2), Weather, Traffic, and Electricity (Wu et al., 2021), with a prediction horizon of $H \in \{96, 192, 336, 720\}$. For the baseline methods, we consider Transformer-based models, including PatchTST (Nie et al., 2023), SimMTM (Dong et al., 2023), FEDformer (Zhou et al., 2022), and Autoformer (Wu et al., 2021), and linear/MLP models, including DLinear (Zeng et al., 2023) and TSMixer (Chen et al., 2023). We also compare PITS and PatchTST without self-supervised pretraining [3], which essentially compares PI and PD architectures only. We follow the experimental setups and baseline results from PatchTST, SimMTM, and TSMixer. For all hyperparameter tuning, we utilize a separate validation dataset, following the standard protocol of splitting all datasets into training, validation, and test sets in chronological order with a ratio of 6:2:2 for the ETT datasets and 7:1:2 for the other datasets (Wu et al., 2021).

**Standard setting.** Table 2 shows the comprehensive results on the multivariate TSF task, demonstrating that our proposed PITS is competitive to PatchTST in both settings, which is the SOTA Transformer-based method, while PITS is much more efficient than PatchTST. SimMTM is a concurrent work showing similar performance to ours in SSL while significantly worse in supervised

---

[3]For PITS and PatchTST supervised learning, patches are overlapped following Nie et al. (2023).

| Models | Self-supervised | | | | | | | | Supervised | | | | | | | | | | | | | |
|---|---|---|---|---|---|---|---|---|---|---|---|---|---|---|---|---|---|---|---|---|---|---|
| | PITS | | PITS w/o CL | | PatchTST* | | SimMTM† | | PITS | | PatchTST | | SimMTM† | | DLinear | | TSMixer | | FEDformer | | Autoformer | |
| Metric | MSE | MAE | MSE | MAE | MSE | MAE | MSE | MAE | MSE | MAE | MSE | MAE | MSE | MAE | MSE | MAE | MSE | MAE | MSE | MAE | MSE | MAE |
| **ETTh1** 96 | 0.367 | 0.393 | 0.367 | 0.393 | 0.379 | 0.408 | 0.367 | 0.402 | 0.369 | 0.397 | 0.375 | 0.399 | 0.380 | 0.412 | 0.375 | 0.399 | 0.361 | 0.392 | 0.376 | 0.415 | 0.435 | 0.446 |
| 192 | 0.401 | 0.416 | 0.400 | 0.413 | 0.414 | 0.428 | 0.403 | 0.425 | 0.403 | 0.416 | 0.414 | 0.421 | 0.416 | 0.434 | 0.405 | 0.416 | 0.404 | 0.418 | 0.423 | 0.446 | 0.456 | 0.457 |
| 336 | 0.415 | 0.428 | 0.425 | 0.430 | 0.435 | 0.446 | 0.415 | 0.430 | 0.409 | 0.426 | 0.431 | 0.436 | 0.448 | 0.458 | 0.439 | 0.443 | 0.420 | 0.431 | 0.444 | 0.462 | 0.486 | 0.487 |
| 720 | 0.425 | 0.452 | 0.444 | 0.459 | 0.468 | 0.474 | 0.430 | 0.453 | 0.456 | 0.465 | 0.449 | 0.466 | 0.481 | 0.469 | 0.472 | 0.490 | 0.463 | 0.472 | 0.469 | 0.492 | 0.515 | 0.517 |
| **ETTh2** 96 | 0.269 | 0.333 | 0.269 | 0.334 | 0.306 | 0.351 | 0.288 | 0.347 | 0.281 | 0.343 | 0.274 | 0.336 | 0.325 | 0.374 | 0.289 | 0.353 | 0.274 | 0.341 | 0.332 | 0.374 | 0.332 | 0.368 |
| 192 | 0.329 | 0.371 | 0.332 | 0.375 | 0.361 | 0.392 | 0.346 | 0.385 | 0.345 | 0.383 | 0.339 | 0.379 | 0.400 | 0.424 | 0.383 | 0.418 | 0.339 | 0.385 | 0.407 | 0.446 | 0.426 | 0.434 |
| 336 | 0.356 | 0.397 | 0.362 | 0.400 | 0.405 | 0.427 | 0.363 | 0.401 | 0.334 | 0.389 | 0.331 | 0.380 | 0.405 | 0.433 | 0.448 | 0.465 | 0.361 | 0.406 | 0.400 | 0.447 | 0.477 | 0.479 |
| 720 | 0.383 | 0.425 | 0.385 | 0.428 | 0.419 | 0.446 | 0.396 | 0.431 | 0.389 | 0.430 | 0.379 | 0.422 | 0.451 | 0.475 | 0.605 | 0.551 | 0.445 | 0.470 | 0.412 | 0.469 | 0.453 | 0.490 |
| **ETTm1** 96 | 0.294 | 0.354 | 0.303 | 0.351 | 0.294 | 0.345 | 0.289 | 0.343 | 0.295 | 0.346 | 0.290 | 0.342 | 0.296 | 0.346 | 0.299 | 0.343 | 0.285 | 0.339 | 0.326 | 0.390 | 0.510 | 0.492 |
| 192 | 0.321 | 0.373 | 0.338 | 0.371 | 0.327 | 0.369 | 0.323 | 0.369 | 0.330 | 0.369 | 0.332 | 0.369 | 0.333 | 0.374 | 0.335 | 0.365 | 0.327 | 0.365 | 0.365 | 0.415 | 0.514 | 0.495 |
| 336 | 0.359 | 0.388 | 0.365 | 0.384 | 0.364 | 0.390 | 0.349 | 0.385 | 0.360 | 0.388 | 0.366 | 0.392 | 0.370 | 0.398 | 0.369 | 0.386 | 0.356 | 0.382 | 0.392 | 0.425 | 0.510 | 0.492 |
| 720 | 0.396 | 0.414 | 0.420 | 0.415 | 0.409 | 0.415 | 0.399 | 0.418 | 0.416 | 0.421 | 0.420 | 0.424 | 0.427 | 0.431 | 0.425 | 0.421 | 0.419 | 0.414 | 0.446 | 0.458 | 0.527 | 0.493 |
| **ETTm2** 96 | 0.165 | 0.260 | 0.160 | 0.253 | 0.167 | 0.256 | 0.166 | 0.257 | 0.163 | 0.255 | 0.165 | 0.255 | 0.175 | 0.268 | 0.167 | 0.260 | 0.163 | 0.252 | 0.180 | 0.271 | 0.205 | 0.293 |
| 192 | 0.213 | 0.291 | 0.213 | 0.291 | 0.232 | 0.302 | 0.223 | 0.295 | 0.215 | 0.293 | 0.220 | 0.292 | 0.240 | 0.312 | 0.224 | 0.303 | 0.216 | 0.290 | 0.252 | 0.318 | 0.278 | 0.336 |
| 336 | 0.263 | 0.325 | 0.263 | 0.325 | 0.291 | 0.342 | 0.282 | 0.334 | 0.266 | 0.329 | 0.278 | 0.329 | 0.298 | 0.351 | 0.281 | 0.342 | 0.268 | 0.324 | 0.324 | 0.364 | 0.343 | 0.379 |
| 720 | 0.337 | 0.373 | 0.339 | 0.375 | 0.368 | 0.390 | 0.370 | 0.385 | 0.342 | 0.380 | 0.367 | 0.385 | 0.403 | 0.413 | 0.397 | 0.421 | 0.420 | 0.422 | 0.410 | 0.420 | 0.414 | 0.419 |
| **Weather** 96 | 0.151 | 0.201 | 0.154 | 0.205 | 0.146 | 0.194 | 0.151 | 0.202 | 0.154 | 0.202 | 0.152 | 0.199 | 0.166 | 0.216 | 0.176 | 0.237 | 0.145 | 0.198 | 0.238 | 0.314 | 0.249 | 0.329 |
| 192 | 0.195 | 0.242 | 0.200 | 0.247 | 0.192 | 0.238 | 0.223 | 0.295 | 0.191 | 0.242 | 0.197 | 0.243 | 0.208 | 0.254 | 0.220 | 0.282 | 0.191 | 0.242 | 0.275 | 0.329 | 0.325 | 0.370 |
| 336 | 0.244 | 0.280 | 0.245 | 0.282 | 0.245 | 0.280 | 0.246 | 0.283 | 0.245 | 0.280 | 0.249 | 0.283 | 0.257 | 0.290 | 0.265 | 0.319 | 0.242 | 0.280 | 0.339 | 0.377 | 0.351 | 0.391 |
| 720 | 0.314 | 0.330 | 0.312 | 0.330 | 0.320 | 0.336 | 0.320 | 0.338 | 0.309 | 0.330 | 0.320 | 0.335 | 0.326 | 0.338 | 0.323 | 0.362 | 0.320 | 0.336 | 0.389 | 0.409 | 0.415 | 0.426 |
| **Traffic** 96 | 0.372 | 0.258 | 0.374 | 0.266 | 0.393 | 0.275 | 0.368 | 0.262 | 0.375 | 0.264 | 0.367 | 0.251 | 0.471 | 0.309 | 0.410 | 0.282 | 0.376 | 0.264 | 0.576 | 0.359 | 0.597 | 0.371 |
| 192 | 0.396 | 0.271 | 0.395 | 0.270 | 0.376 | 0.254 | 0.373 | 0.251 | 0.389 | 0.270 | 0.385 | 0.259 | 0.475 | 0.308 | 0.423 | 0.287 | 0.397 | 0.264 | 0.610 | 0.380 | 0.607 | 0.382 |
| 336 | 0.411 | 0.280 | 0.408 | 0.277 | 0.384 | 0.259 | 0.395 | 0.254 | 0.401 | 0.277 | 0.398 | 0.265 | 0.490 | 0.315 | 0.436 | 0.296 | 0.413 | 0.290 | 0.608 | 0.375 | 0.623 | 0.387 |
| 720 | 0.436 | 0.290 | 0.436 | 0.290 | 0.446 | 0.306 | 0.432 | 0.290 | 0.437 | 0.294 | 0.434 | 0.287 | 0.524 | 0.332 | 0.466 | 0.315 | 0.444 | 0.306 | 0.621 | 0.375 | 0.639 | 0.395 |
| **Electricity** 96 | 0.130 | 0.225 | 0.131 | 0.226 | 0.126 | 0.221 | 0.133 | 0.223 | 0.132 | 0.228 | 0.130 | 0.222 | 0.190 | 0.279 | 0.140 | 0.237 | 0.131 | 0.229 | 0.186 | 0.302 | 0.196 | 0.313 |
| 192 | 0.144 | 0.240 | 0.145 | 0.240 | 0.145 | 0.238 | 0.147 | 0.237 | 0.147 | 0.242 | 0.148 | 0.240 | 0.195 | 0.285 | 0.153 | 0.249 | 0.151 | 0.246 | 0.197 | 0.311 | 0.211 | 0.324 |
| 336 | 0.160 | 0.256 | 0.162 | 0.256 | 0.164 | 0.256 | 0.166 | 0.265 | 0.162 | 0.261 | 0.167 | 0.261 | 0.211 | 0.301 | 0.169 | 0.267 | 0.161 | 0.261 | 0.213 | 0.328 | 0.214 | 0.327 |
| 720 | 0.194 | 0.287 | 0.201 | 0.290 | 0.200 | 0.290 | 0.203 | 0.297 | 0.199 | 0.290 | 0.202 | 0.291 | 0.253 | 0.333 | 0.203 | 0.301 | 0.197 | 0.293 | 0.233 | 0.344 | 0.236 | 0.342 |
| Average | 0.301 | 0.327 | 0.304 | 0.328 | 0.314 | 0.333 | 0.306 | 0.331 | 0.304 | 0.329 | 0.307 | 0.327 | 0.343 | 0.355 | 0.332 | 0.351 | 0.311 | 0.333 | 0.373 | 0.386 | 0.412 | 0.409 |

\* We used the official code to replicate the results.  † SimMTM is a concurrent work to ours.

Table 2: **Results of multivariate TSF.** We compare both the supervised and self-supervised versions of PatchTST and our method. The best results are in bold and the second best are underlined.

| Metric: MSE | PITS | | | PatchTST | | |
|---|---|---|---|---|---|---|
| | FT | LP | Sup. | FT | LP | Sup. |
| ETTh1 | 0.401 | 0.403 | 0.409 | 0.424 | 0.434 | 0.417 |
| ETTh2 | 0.334 | 0.335 | 0.337 | 0.373 | 0.364 | 0.331 |
| ETTm1 | 0.342 | 0.356 | 0.351 | 0.349 | 0.355 | 0.352 |
| ETTm2 | 0.244 | 0.244 | 0.247 | 0.264 | 0.264 | 0.258 |
| Weather | 0.225 | 0.239 | 0.225 | 0.226 | 0.233 | 0.230 |
| Traffic | 0.403 | 0.356 | 0.401 | 0.401 | 0.424 | 0.396 |
| Electricity | 0.157 | 0.161 | 0.160 | 0.159 | 0.168 | 0.162 |
| Average | 0.301 | 0.306 | 0.304 | 0.314 | 0.320 | 0.307 |

Table 3: PITS vs. PatchTST.

| | Source | Target | PITS | | PatchTST | | SimMTM | TimeMAE | TST | LaST | TF-C | CoST |
|---|---|---|---|---|---|---|---|---|---|---|---|---|
| | | | FT | LP | FT | LP | | | | | | |
| In-domain | ETTh2 | ETTh1 | 0.404 | 0.403 | 0.423 | 0.464 | 0.415 | 0.728 | 0.645 | 0.443 | 0.635 | 0.584 |
| | ETTm2 | ETTm1 | 0.345 | 0.354 | 0.348 | 0.411 | 0.351 | 0.682 | 0.480 | 0.414 | 0.750 | 0.354 |
| | Average | | 0.375 | 0.378 | 0.386 | 0.406 | 0.383 | 0.705 | 0.563 | 0.429 | 0.697 | 0.469 |
| Cross-domain | ETTm2 | ETTh1 | 0.407 | 0.405 | 0.433 | 0.421 | 0.428 | 0.724 | 0.632 | 0.503 | 1.091 | 0.582 |
| | ETTh2 | ETTm1 | 0.350 | 0.357 | 0.363 | 0.378 | 0.365 | 0.688 | 0.472 | 0.475 | 0.750 | 0.377 |
| | ETTm1 | ETTh1 | 0.409 | 0.409 | 0.447 | 0.432 | 0.422 | 0.726 | 0.645 | 0.426 | 0.700 | 0.750 |
| | ETTh1 | ETTm1 | 0.352 | 0.357 | 0.348 | 0.374 | 0.346 | 0.666 | 0.482 | 0.353 | 0.746 | 0.359 |
| | Weather | ETTh1 | 0.406 | 0.406 | 0.437 | 0.423 | 0.456 | - | - | - | - | - |
| | Weather | ETTm1 | 0.350 | 0.356 | 0.348 | 0.355 | 0.358 | - | - | - | - | - |
| | Average | | 0.379 | 0.382 | 0.396 | 0.397 | 0.396 | - | - | - | - | - |

Table 4: Results of TSF with transfer learning.

learning. Table 3 compares PITS and PatchTST under three different scenarios: fine-tuning (FT), linear probing (LP), and supervised learning without self-supervised pretraining (Sup.), where we present the average MSE across four horizons. As shown in Table 3, PITS outperforms PatchTST for all scenarios on average.

**Transfer learning.** In in-domain transfer, we experiment datasets with the same frequency for the source and target datasets, whereas in cross-domain transfer, datasets with different frequencies are utilized for the source and target datasets. Table 4 shows the results of the average MSE across four horizons, which demonstrates that our proposed PITS surpasses the SOTA methods in most cases.

## 4.3 TIME SERIES CLASSIFICATION

**Datasets and baseline methods.** For classification tasks, we use five datasets, SleepEEG (Kemp et al., 2000), Epilepsy (Andrzejak et al., 2001), FD-B (Lessmeier et al., 2016), Gesture (Liu et al., 2009), and EMG (Goldberger et al., 2000). For the baseline methods, we employ TS-SD (Shi et al., 2021), TS2Vec (Yue et al., 2022), CoST (Woo et al., 2022), LaST (Wang et al., 2022), Mixing-Up (Wickstrøm et al., 2022), TS-TCC (Eldele et al., 2021), TF-C (Zhang et al., 2022), TST (Zerveas et al., 2021), TimeMAE (Cheng et al., 2023) and SimMTM (Dong et al., 2023).

**Standard setting.** Table 5 demonstrates that our proposed PITS outperforms all SOTA methods in all metrics on the SleepEEG dataset. This contrasts with the results in prior works that CL is superior to MTM for classification tasks (Yue et al., 2022): while prior MTM methods such as TST and TimeMAE shows relatively low performance compared to CL methods such as TS2Vec and TF-C[4], the proposed PITS outperforms CL methods, even without complementary CL.

**Transfer learning.** For transfer learning, we conduct experiments in both in-domain and cross-domain transfer settings, using SleepEEG as the source dataset for both settings. For in-domain transfer, we use target datasets from the same domain as the source dataset, which share the characteristic of

---

[4]An exception is SimMTM (Dong et al., 2023), which is not officially published at the time of submission.

| | ACC. | PRE. | REC. | $F_1$ |
|---|---|---|---|---|
| TS2Vec | 92.17 | 93.84 | 81.19 | 85.71 |
| CoST | 88.07 | 91.58 | 66.05 | 69.11 |
| LaST | 92.11 | 93.12 | 81.47 | 85.74 |
| TF-C | 93.96 | 94.87 | 85.82 | 89.46 |
| TST | 80.21 | 40.11 | 50.00 | 44.51 |
| TimeMAE | 80.34 | 90.16 | 50.33 | 45.20 |
| SimMTM | 94.75 | 95.60 | 89.93 | 91.41 |
| PITS w/o CL | 95.27 | 95.35 | 95.27 | 95.30 |
| PITS | **95.67** | **95.63** | **95.67** | **95.64** |

Table 5: Results of TSC.

| | In-domain transfer learning | | | | | | | | Cross-domain transfer learning | | | | | | | |
|---|---|---|---|---|---|---|---|---|---|---|---|---|---|---|---|---|
| | SleepEEG → Epilepsy | | | | SleepEEG → FD-B | | | | SleepEEG → Gesture | | | | SleepEEG → EMG | | | |
| | ACC. | PRE. | REC. | $F_1$ | ACC. | PRE. | REC. | $F_1$ | ACC. | PRE. | REC. | $F_1$ | ACC. | PRE. | REC. | $F_1$ |
| TS-SD | 89.52 | 80.18 | 76.47 | 77.67 | 55.66 | 57.10 | 60.54 | 57.03 | 69.22 | 66.98 | 68.67 | 66.56 | 46.06 | 15.45 | 33.33 | 21.11 |
| TS2Vec | 93.95 | 90.59 | 90.39 | 90.45 | 47.90 | 43.39 | 48.42 | 43.89 | 69.17 | 65.45 | 68.54 | 65.70 | 78.54 | 80.40 | 67.85 | 67.66 |
| CoST | 88.40 | 88.20 | 72.34 | 76.88 | 47.06 | 38.79 | 38.42 | 34.79 | 68.33 | 65.30 | 68.33 | 66.42 | 53.65 | 49.07 | 42.10 | 35.27 |
| LaST | 86.46 | 90.77 | 66.35 | 70.67 | 46.67 | 43.90 | 47.71 | 45.17 | 64.17 | 70.36 | 64.17 | 58.76 | 66.34 | 79.34 | 63.33 | 72.55 |
| Mixing-Up | 80.21 | 40.11 | 50.00 | 44.51 | 67.89 | 71.46 | 76.13 | 72.73 | 69.33 | 67.19 | 69.33 | 64.97 | 30.24 | 10.99 | 25.83 | 15.41 |
| TS-TCC | 92.53 | 94.51 | 81.81 | 86.33 | 54.99 | 52.79 | 63.96 | 54.18 | 71.88 | 71.35 | 71.67 | 69.84 | 78.89 | 58.51 | 63.10 | 59.04 |
| TF-C | 94.95 | 94.56 | 89.08 | 91.49 | 69.38 | 75.59 | 72.02 | 74.87 | 76.42 | 77.31 | 74.29 | 75.72 | 81.71 | 72.65 | 81.59 | 76.83 |
| TST | 80.21 | 40.11 | 50.00 | 44.51 | 46.40 | 41.58 | 45.50 | 41.34 | 69.17 | 66.60 | 69.17 | 66.01 | 46.34 | 15.45 | 33.33 | 21.11 |
| TimeMAE | 89.71 | 72.36 | 67.47 | 68.55 | 70.88 | 66.98 | 68.94 | 66.56 | 71.88 | 70.35 | 76.75 | 68.37 | 69.99 | 70.25 | 63.44 | 70.89 |
| SimMTM | 95.49 | 93.36 | 92.28 | 92.81 | 69.40 | 74.18 | 76.41 | 75.11 | 80.00 | 79.03 | 80.00 | 78.67 | 97.56 | 98.33 | 98.04 | 98.14 |
| PITS | **95.71** | **95.69** | **95.71** | **95.70** | **88.65** | **88.86** | **88.65** | **88.63** | **92.50** | **93.32** | **92.50** | **92.48** | **100.0** | **100.0** | **100.0** | **100.0** |

Table 6: Results of TSC with transfer learning.

| | PI acrhitecture | | | | | | PD architecture | | | | | |
|---|---|---|---|---|---|---|---|---|---|---|---|---|
| | Linear | | | MLP | | | MLP-Mixer | | | Transformer | | |
| Task | PD | PI | Gain(%) | PD | PI | Gain(%) | PD | PI | Gain(%) | PD | PI | Gain(%) |
| ETTh1 | 0.408 | 0.408 | +0.0 | 0.418 | 0.407 | **+2.6** | 0.420 | 0.409 | **+2.6** | 0.425 | 0.415 | **+2.4** |
| ETTh2 | 0.343 | 0.338 | **+1.5** | 0.361 | 0.334 | **+7.5** | 0.365 | 0.341 | **+6.6** | 0.353 | 0.342 | **+3.1** |
| ETTm1 | 0.359 | 0.358 | **+0.2** | 0.356 | 0.355 | **+0.3** | 0.354 | 0.352 | **+0.6** | 0.350 | 0.350 | +0.0 |
| ETTm2 | 0.254 | 0.243 | **+0.4** | 0.258 | 0.253 | **+1.9** | 0.259 | 0.253 | **+2.3** | 0.274 | 0.256 | **+6.6** |
| Average | 0.342 | 0.340 | **+0.3** | 0.348 | 0.337 | **+3.2** | 0.350 | 0.339 | **+3.1** | 0.351 | 0.341 | **+2.8** |

Table 7: **Effectiveness of PI strategies.** Pretraining with the PI task consistently outperforms the PD task across all architectures. The results are reported as the average across four horizons.

being EEG datasets, while we use target datasets from the different domain for cross-domain transfer. Table 6 demonstrates that our PITS outperforms SOTA methods in all scenarios. In particular, the performance gain is significant in the challenging cross-domain transfer learning setting, implying that PITS would be more practical in real-world applications under domain shifts.

## 4.4 ABLATION STUDY

**Effect of PI/PD tasks/architectures.** To assess the effect of our proposed PI pretraining task and PI encoder architecture, we conduct an ablation study in Table 7 using a common input horizon of 512 and patch size of 12. Recall that the PD task predicts masked patches using unmasked patches while the PI task autoencodes patches, and the PD architectures include interaction among patches using either the fully-connected layer (MLP-Mixer) or the self-attention module (Transformer), while the PI architectures (Linear, MLP) do not. As shown in Table 7, PI pretraining results in better TSF performance than PD pretraining regardless of the choice of the architecture. Also, PI architectures exhibit competitive performance compared to PD architectures, while PI architectures are more lightweight and efficient as demonstrated in Table 13. Among them, MLP shows the best performance while keeping efficiency, so we use MLP as the architecture of PITS throughout all experiments.

**Hidden dimension and dropout.** The PI task may raise a concern on the trivial solution: when the hidden dimension $D$ is larger than the input dimension $P$, the identity mapping perfectly reconstructs the input. This can be addressed by introducing dropout, where we add a dropout layer before the linear projection head. Figure 4 displays the average MSE on four ETT datasets across four horizons under various hidden dimensions $D$ in MLP with a common input horizon of 512, without dropout or with the dropout rate of 0.2. Note that for this experiment, the input dimension (patch size) is 12, and a trivial solution can occur if $D \geq 12$. The results confirm that using dropout is necessary to learn high

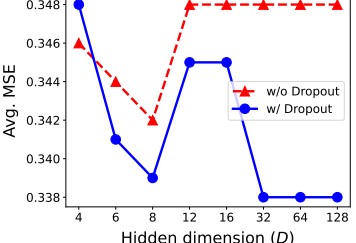

Figure 4: MSE by $D$ and dropout.

dimensional representations, leading to better performance. Based on this result, we tune $D \in \{32, 64, 128\}$ throughout experiments, while performance is consistent with $D$ values in the range. An ablation study with different dropout rates can be found in Appendix K.

**Performance of various pretrain tasks.** In addition to the 1) PD task of reconstructing the masked patches ($X_m$) and 2) PI task of autoencoding the unmasked patches ($X_u$), we also employ two other basic tasks for comparison: 3) predicting $X_u$ from zero-filled patches (**0**) and 4) autoencoding **0**. Table 8 displays the average MSE on four ETT datasets across four horizons with a common input horizon of 512, highlighting that the model pretrained with the PD task performs even worse than

| Pretrain Task | | Trans-former | MLP | |
|---|---|---|---|---|
| Input | Output | | w/o CL | w CL |
| $X_u$ | $X_u$ | **0.341** | **0.338** | **0.330** |
| $X_u$ | $X_m$ | 0.351 | 0.348 | 0.364 |
| **0** | $X_u$ | 0.342 | 0.348 | 0.348 |
| **0** | **0** | 0.343 | 0.345 | 0.345 |

Table 8: Pretraining tasks.

| Layer 1 | - | - | - | PI | CL |
|---|---|---|---|---|---|
| Layer 2 | CL | PI | CL+PI | CL | PI |
| ETTh1 | 0.720 | 0.407 | 0.417 | 0.442 | **0.401** |
| ETTh2 | 0.394 | 0.334 | 0.366 | 0.371 | **0.334** |
| ETTm1 | 0.711 | 0.357 | 0.356 | 0.358 | **0.342** |
| ETTm2 | 0.381 | 0.253 | 0.254 | 0.265 | **0.244** |
| Avg. | 0.552 | 0.338 | 0.348 | 0.359 | **0.330** |

Table 9: Effect of CL.

| | $z_1$ | $z_2$ | $z_2^*$ |
|---|---|---|---|
| 96 | 0.371 | **0.367** | 0.369 |
| 192 | **0.396** | 0.401 | 0.403 |
| 336 | **0.411** | 0.415 | 0.428 |
| 720 | 0.448 | **0.425** | 0.460 |
| Avg. | 0.407 | **0.401** | 0.415 |

Table 10: Representation for downstream tasks.

| PI task | | ETTh1 | ETTh2 | ETTm1 | ETTm2 | Avg. |
|---|---|---|---|---|---|---|
| Transformer | | 0.425 | 0.353 | 0.350 | 0.274 | 0.351 |
| MLP | w/o CL | 0.407 | 0.334 | 0.357 | 0.253 | 0.338 |
| | w/ non-hier. CL | 0.405 | 0.333 | 0.353 | 0.252 | 0.336 |
| | w/ hier. CL | **0.401** | **0.334** | **0.342** | **0.244** | **0.330** |

Table 11: Hierarchical design of complementary CL.

| 1) Encoder Architecture | | |
|---|---|---|
| Transformer | Linear | MLP |
| 0.425* | 0.408 | 0.418 |
| 2) PD task → PI task | | |
| 0.415 | 0.408 | 0.407 |
| 3) + Complementary CL | | |
| - | - | **0.401** |

Table 12: PatchTST→PITS

the two basic tasks with **0** as inputs. This emphasizes the ineffectiveness of the PD task and the effectiveness of the proposed PI task.

**Which representation to use for downstream tasks?** In SSL, the boundary of the encoder and the task-specific projection head is often unclear. To determine the location to extract representation for downstream tasks, we conduct experiments using representations from intermediate layers in MLP: 1) $z_1$ from the first layer, 2) $z_2$ from the second layer, and 3) $z_2^*$ from the additional projection layer attached on top of the second layer. Table 10 displays the MSE of ETTh1 across four horizons, indicating that the second layer $z_2$ yields the best results.

**Location of complementary CL.** To assess the effect of complementary CL together with PI reconstruction, we conduct an ablation study on the choice of pretext tasks and their location in the MLP encoder: the contrastive and/or reconstruction loss is computed on the first or second layer, or neither. Table 9 displays the average MSE on four ETT datasets across four horizons. We observe that the PI reconstruction task is essential, and CL is effective when it is considered in the first layer.

**Hierarchical design of complementary CL.** The proposed complementary CL is structured hierarchically to capture both coarse and fine-grained information in time series. To evaluate the effect of this hierarchical design, we consider three different options: 1) without CL, 2) with non-hierarchical CL, and 3) with hierarchical CL. Table 11 presents the average MSE on four ETT datasets across four horizons, highlighting the performance gain by the hierarchical design.

**Comparison with PatchTST.** PITS can be derived from PatchTST, by changing the pretraining task and encoder architecture. Table 12 shows how each modification contributes to the performance improvement on the ETTh1 dataset. Note that we apply mask ratio of 50% to PatchTST, which does not affect the performance (marked with *).

## 5 ANALYSIS

**PI task is more robust to distribution shift than PD task.** To assess the robustness of pretraining tasks to distribution shifts, which are commonly observed in real-world datasets (Han et al., 2023), we generate 98 toy examples exhibiting varying degrees of distribution shift, as depicted in the left panel of Figure 5. The degree of shift is characterized by changes in slope and amplitude. The right panel of Figure 5 visualizes the performance gap between the models trained with the PD and PI tasks, where the horizontal and vertical axis correspond to the slope and amplitude differences between training and test phases, respectively. The result indicates that the model trained with the PI task exhibits overall better robustness to distribution shifts as the MSE difference is non-negative in all regime and the gap increases as the shift becomes more severe, particularly when the slope is flipped or amplitude is increased.

**MLP is more robust to patch size than Transformer.** To assess the robustness of encoder architectures to patch size, we compare MLP and Transformer using ETTh1 with different patch sizes. Figure 6 illustrates the results, indicating that MLP is more robust for both the PI and PD tasks, resulting in consistently better forecasting performance across various patch sizes.

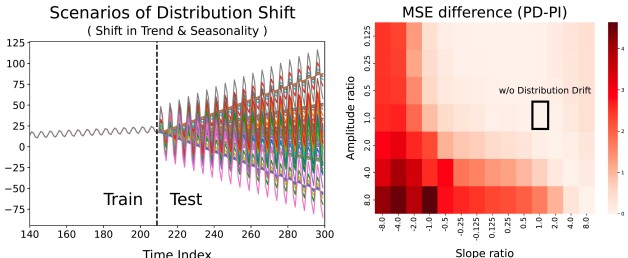

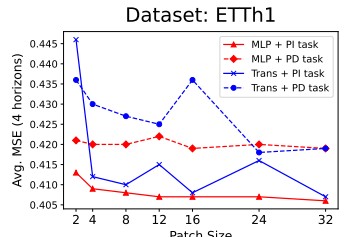

Figure 5: PI vs. PD tasks under distribution shifts.

Figure 6: MSE by patch size.

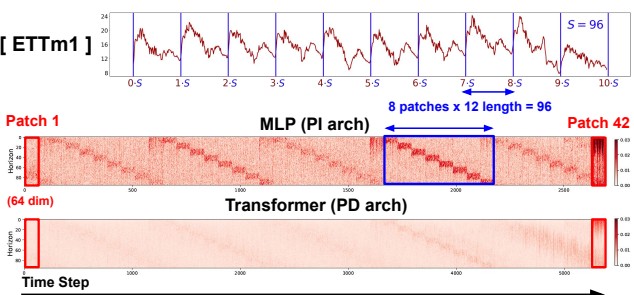

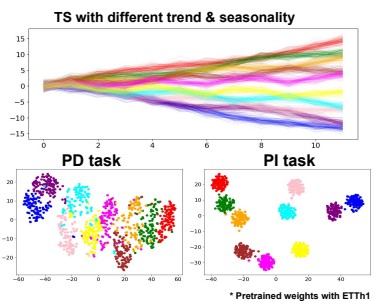

Figure 7: Downstream task weight $W \in \mathbb{R}^{H \times N \cdot D}$.

Figure 8: t-SNE visualization.

**MLP is more interpretable than Transformer.** While PI architectures process each patch independently, PD architectures share information from all patches, leading to information leaks among patches. This makes MLP more interpretable than Transformer, as visualizing the weight matrix of the linear layer additionally introduced and learned for the downstream task shows each patch's contribution to predictions. Figure 7 illustrates the seasonality of ETTm1 and the downstream weight matrix trained on ETTm1 for both architectures. While the weight matrix of the linear layer on top of Transformer is mostly uniform, that of MLP reveals seasonal patterns and emphasizes recent information, highlighting that MLP captures the seasonality better than Transformer.

**Efficiency analysis.** To demonstrate the efficiency of the PI architecture, we compare PatchTST and PITS in terms of the number of parameters and training/inference time on ETTm2. As shown in Table 13, PITS outperforms PatchTST with significantly fewer parameters and faster training and inference, where we pretrain for 100 epochs and perform inference with the entire test dataset. The comparison of the efficiency between self-supervised and supervised settings is provided in Appendix J.

|  | Self-supervised settings | | | |
|---|---|---|---|---|
|  | PatchTST | PITS | | |
|  |  | w/o CL | w/ CL | w/ hier. CL |
| Number of params | 406,028 | 5,772 | | |
| Pretrain time (min) | 77 | 15 | 17 | 25 |
| Inference time (sec) | 7.5 | 3.3 | | |
| Avg. MSE | 0.274 | 0.253 | 0.252 | 0.244 |

Table 13: Time/parameter efficiency.

**t-SNE visualization.** To evaluate the quality of representations obtained from the PI and PD tasks, we utilize t-SNE (Van der Maaten & Hinton, 2008) for visualization. For this analysis, we create toy examples with 10 classes of its own trend and seasonality patterns, as shown in Figure 8. The results demonstrate that representations learned from the PI task better distinguishes between classes.

## 6  CONCLUSION

This paper revisits masked modeling in time series analysis, focusing on two key aspects: 1) the pretraining task and 2) the model architecture. In contrast to previous works that primarily emphasize dependencies between TS patches, we advocate a patch-independent approach on two fronts: 1) by introducing a patch reconstruction task and 2) employing patch-wise MLP. Our results demonstrate that the proposed PI approach is more robust to distribution shifts and patch size compared to the PD approach, resulting in superior performance while more efficient in both forecasting and classification tasks. We hope that our work sheds light on the effectiveness of self-supervised learning through simple pretraining tasks and model architectures in various domains, and provides a strong baseline to future works on time series analysis.

## ETHICS STATEMENT

The proposed self-supervised learning algorithm, employing patch-independent strategies in terms of pretraining tasks and model architecture, holds the potential to have a significant impact in the field of representation learning for time series, especially in scenarios where annotation is scarce or not available. This algorithm can be effectively applied in various real-world settings, encompassing both forecasting and classification tasks, even in situations where distribution shifts are severe. Furthermore, we foresee that the concept of utilizing lightweight architectures will serve as a source of inspiration for future endeavors across domains where substantial computational resources are not readily accessible.

Nevertheless, as is the case with any algorithm, ethical considerations come to the forefront. One notable ethical concern relates to the possibility of the algorithm perpetuating biases inherent in the pretraining datasets. It is necessary to assess and mitigate potential biases within the pretraining dataset before deploying the algorithm in real-world applications. To ensure the responsible utilization of the algorithm, we are committed to providing the source code which will promote transparency and reproducibility, enabling fellow researchers to scrutinize and rectify potential biases and guard against any misuse.

## ACKNOWLEDGEMENTS

This work was supported by the National Research Foundation of Korea (NRF) grant funded by the Korea government (MSIT) (2020R1A2C1A01005949, 2022R1A4A1033384, RS-2023-00217705), the MSIT(Ministry of Science and ICT), Korea, under the ICAN(ICT Challenge and Advanced Network of HRD) support program (RS-2023-00259934) supervised by the IITP(Institute for Information & Communications Technology Planning & Evaluation), the Yonsei University Research Fund (2023-22-0071), and the Son Jiho Research Grant of Yonsei University (2023-22-0006).

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

# A    DATASET DESCRIPTION

## A.1    TIME SERIES FORECASTING

For time series forecasting, we assess the effectiveness of our proposed PITS using seven datasets, including four ETT datasets (ETTh1, ETTh2, ETTm1, ETTm2), Weather, Traffic, and Electricity. These datasets have been widely employed for benchmarking and are publicly accessible (Wu et al., 2021). The statistics of these datasets are summarized in Table A.1.

| Datasets | ETTh1 | ETTh2 | ETTm1 | ETTm2 | Weather | Traffic | Electricity |
|---|---|---|---|---|---|---|---|
| Features | 7 | 7 | 7 | 7 | 21 | 862 | 321 |
| Timesteps | 17420 | 17420 | 69680 | 69680 | 52696 | 17544 | 26304 |

Table A.1: Statistics of datasets for forecasting.

## A.2    TIME SERIES CLASSIFICATION

For time series classification, we use five datasets of different characteristics, as described in Table A.2. Note that both SleepEEG and Epilepsy datasets belong to the same domain, characterized by being EEG datasets. For transfer learning tasks, we define them as being part of the same domain.

| Dataset | # Samples | # Channels | # Classes | Length | Freq (Hz) |
|---|---|---|---|---|---|
| SleepEEG | 371,055 | 1 | 5 | 200 | 100 |
| Epilepsy | 60 / 20 / 11,420 | 1 | 2 | 178 | 174 |
| FD-B | 60 / 21 / 13,559 | 1 | 3 | 5,120 | 64,000 |
| Gesture | 320 / 120 / 120 | 3 | 8 | 315 | 100 |
| EMG | 122 / 41 / 41 | 1 | 3 | 1,500 | 4,000 |

Table A.2: Statistics of datasets for classification.

# B    EXPERIMENTAL SETTINGS

We follow the standard practice of splitting all datasets into training, validation, and test sets in chronological order (Wu et al., 2021). The splitting ratios were set at 6:2:2 for the ETT dataset and 7:1:2 for the other datasets. It is important to note that we benefit from minimal hyperparameters due to our use of a simple architecture. We conduct hyperparameter search for three key parameters using the predefined validation dataset: the hidden dimension of the MLP ($D \in \{32, 64, 128\}$), patch size ($P \in \{12, 18, 24\}$), and input horizon ($L \in 336, 512, 768$). For self-supervised learning, we utilize a shared pretrained weight for all prediction horizons, making it more efficient compared to supervised learning in the long term.

In both self-supervised pretraining and supervised learning, we utilize an epoch size of 100. During fine-tuning in self-supervised learning, we apply linear probing for either 10 or 20 epochs, depending on the dataset, to update the model head. Subsequently, we perform end-to-end fine-tuning of the entire network for twice the epoch duration of linear probing, following the approach outlined in PatchTST (Nie et al., 2023). The dropout ratio for the fully connected layer preceding the prediction head is set to 0.2.

# C HYPERPARAMETERS

## C.1 TIME SERIES FORECASTING

### C.1.1 SELF-SUPERVISED LEARNING

| Dataset | Architecture | | | Epochs | | |
|---|---|---|---|---|---|---|
| | Dimension ($D$) | Patch size ($P$) | Number of patches ($N$) | Pretrain | Fine-tuning | Linear-probing |
| ETTh1 | 128 | 12 | 42 | 100 | 5 | 5 |
| ETTh2 | 128 | 12 | 42 | 100 | 5 | 5 |
| ETTm1 | 64 | 18 | 42 | 100 | 5 | 20 |
| ETTm2 | 64 | 24 | 42 | 100 | 5 | 20 |
| Weather | 128 | 24 | 32 | 100 | 30 | 20 |
| Traffic | 256 | 24 | 32 | 100 | 20 | 20 |
| Electricity | 256 | 32 | 32 | 100 | 30 | 20 |

### C.1.2 SUPERVISED LEARNING

| Dataset | Architecture | | | Epochs |
|---|---|---|---|---|
| | Dimension ($D$) | Patch size ($P$) | Number of patches ($N$) | |
| ETTh1 | 256 | 24 | 42 | 100 |
| ETTh2 | 64 | 24 | 28 | 100 |
| ETTm1 | 64 | 24 | 28 | 100 |
| ETTm2 | 128 | 24 | 64 | 100 |
| Weather | 128 | 24 | 42 | 100 |
| Traffic | 64 | 24 | 64 | 100 |
| Electricity | 32 | 24 | 64 | 100 |

### C.1.3 TRANSFER LEARNING

| Dataset | | Epochs | |
|---|---|---|---|
| Source | Target | Fine-tuning | Linear-probing |
| ETTh2 | ETTh1 | 5 | 10 |
| ETTm2 | ETTm1 | 20 | 10 |
| ETTm2 | ETTh1 | 10 | 10 |
| ETTh2 | ETTm1 | 5 | 10 |
| ETTh2 | ETTm1 | 5 | 20 |
| ETTm1 | ETTh1 | 5 | 20 |
| ETTh1 | ETTm1 | 5 | 20 |
| Weather | ETTh1 | 10 | 20 |
| Weather | ETTm1 | 5 | 20 |

## C.2  TIME SERIES CLASSIFICATION

### C.2.1  TRANSFER LEARNING

| Dataset | | Architecture | | | | Epochs | |
|---------|--------|--------------|----------------|-----------------------|-----------|----------|-------------|
| Source | Target | Dimension ($D$) | Patch size ($P$) | Number of patches ($N$) | Aggregate | Pretrain | Fine-tuning |
| SleepEEG | Epilepsy | 64 | 8 | 22 | max | 20 | 150 |
| | FD-B | 128 | 8 | 22 | avg | 60 | 2000 |
| | Gesture | 128 | 16 | 11 | concat | 20 | 100 |
| | EMG | 64 | 8 | 22 | max | 100 | 3000 |

## D  TIME SERIES FORECASTING

To demonstrate the effectiveness of PITS compared to other SOTA self-supervised methods, we compare PITS with methods including PatchTST (Nie et al., 2023), SimMTM (Dong et al., 2023), TimeMAE (Cheng et al., 2023), TST (Zerveas et al., 2021) as MTM methods, and TF-C (Zhang et al., 2022), CoST (Woo et al., 2022), TS2Vec (Yue et al., 2022) as CL methods. The results presented in Table D.1 showcase the superior performance of PITS over these methods in multivariate time series forecasting task.

| Models | | \multicolumn{18}{c}{Self-supervised} |
|--------|---|

| Models | | PITS | | PITS w/o CL | | PatchTST* | | SimMTM† | | TimeMAE | | TST | | LaST | | TF-C | | CoST | | TS2Vec | |
|--------|-----|-------|-------|-------|-------|-------|-------|-------|-------|-------|-------|-------|-------|-------|-------|-------|-------|-------|-------|-------|-------|
| Metric | | MSE | MAE | MSE | MAE | MSE | MAE | MSE | MAE | MSE | MAE | MSE | MAE | MSE | MAE | MSE | MAE | MSE | MAE | MSE | MAE |
| ETTh1 | 96 | **0.367** | **0.393** | **0.367** | **0.393** | 0.379 | 0.408 | **0.367** | 0.402 | 0.708 | 0.570 | 0.503 | 0.527 | 0.399 | 0.412 | 0.463 | 0.406 | 0.514 | 0.512 | 0.709 | 0.650 |
| | 192 | 0.401 | 0.416 | **0.400** | **0.413** | 0.414 | 0.428 | 0.403 | 0.425 | 0.725 | 0.587 | 0.601 | 0.552 | 0.484 | 0.468 | 0.531 | 0.540 | 0.655 | 0.590 | 0.927 | 0.757 |
| | 336 | **0.415** | **0.428** | 0.425 | 0.430 | 0.435 | 0.446 | **0.415** | 0.430 | 0.713 | 0.589 | 0.625 | 0.541 | 0.580 | 0.533 | 0.535 | 0.545 | 0.790 | 0.666 | 0.986 | 0.811 |
| | 720 | **0.425** | **0.452** | 0.444 | 0.459 | 0.468 | 0.474 | 0.430 | 0.453 | 0.736 | 0.618 | 0.768 | 0.628 | 0.432 | 0.432 | 0.577 | 0.562 | 0.880 | 0.739 | 0.967 | 0.790 |
| ETTh2 | 96 | **0.269** | **0.333** | **0.269** | 0.334 | 0.306 | 0.351 | 0.288 | 0.347 | 0.443 | 0.465 | 0.335 | 0.392 | 0.331 | 0.390 | 0.463 | 0.521 | 0.465 | 0.482 | 0.506 | 0.477 |
| | 192 | **0.329** | **0.371** | 0.332 | 0.375 | 0.361 | 0.392 | 0.346 | 0.385 | 0.533 | 0.516 | 0.444 | 0.441 | 0.451 | 0.452 | 0.525 | 0.561 | 0.671 | 0.599 | 0.567 | 0.547 |
| | 336 | **0.356** | **0.397** | 0.362 | 0.400 | 0.405 | 0.427 | 0.363 | 0.401 | 0.445 | 0.472 | 0.455 | 0.494 | 0.460 | 0.478 | 0.850 | 0.883 | 0.848 | 0.776 | 0.694 | 0.628 |
| | 720 | **0.383** | **0.425** | 0.385 | 0.428 | 0.419 | 0.446 | 0.396 | 0.431 | 0.507 | 0.498 | 0.455 | 0.504 | 0.552 | 0.509 | 0.930 | 0.932 | 0.871 | 0.811 | 0.728 | 0.838 |
| ETTm1 | 96 | 0.294 | 0.354 | 0.303 | 0.351 | 0.294 | 0.345 | **0.289** | **0.343** | 0.647 | 0.497 | 0.454 | 0.456 | 0.316 | 0.355 | 0.419 | 0.401 | 0.376 | 0.420 | 0.563 | 0.551 |
| | 192 | **0.321** | 0.373 | 0.338 | 0.371 | 0.327 | 0.369 | 0.323 | 0.369 | 0.597 | 0.508 | 0.471 | 0.490 | 0.349 | 0.366 | 0.471 | 0.438 | 0.420 | 0.451 | 0.599 | 0.558 |
| | 336 | **0.359** | **0.388** | 0.365 | 0.384 | 0.364 | 0.390 | 0.349 | 0.385 | 0.699 | 0.525 | 0.457 | 0.451 | 0.429 | 0.407 | 0.540 | 0.509 | 0.482 | 0.494 | 0.685 | 0.594 |
| | 720 | **0.396** | **0.414** | 0.420 | 0.415 | 0.409 | 0.415 | 0.399 | 0.418 | 0.786 | 0.596 | 0.594 | 0.488 | 0.496 | 0.464 | 0.552 | 0.548 | 0.628 | 0.578 | 0.831 | 0.698 |
| ETTm2 | 96 | 0.165 | 0.260 | **0.160** | **0.253** | 0.167 | 0.256 | 0.166 | 0.257 | 0.304 | 0.357 | 0.363 | 0.301 | 0.163 | 0.255 | 0.401 | 0.477 | 0.276 | 0.384 | 0.448 | 0.482 |
| | 192 | **0.213** | **0.291** | **0.213** | **0.291** | 0.232 | 0.302 | 0.223 | 0.295 | 0.334 | 0.387 | 0.342 | 0.364 | 0.239 | 0.303 | 0.422 | 0.490 | 0.500 | 0.532 | 0.545 | 0.536 |
| | 336 | **0.263** | **0.325** | **0.263** | **0.325** | 0.291 | 0.342 | 0.282 | 0.334 | 0.420 | 0.441 | 0.414 | 0.361 | 0.259 | 0.366 | 0.513 | 0.508 | 0.680 | 0.695 | 0.681 | 0.744 |
| | 720 | **0.337** | **0.373** | 0.339 | 0.375 | 0.368 | 0.390 | 0.370 | 0.385 | 0.508 | 0.481 | 0.580 | 0.456 | 0.397 | 0.382 | 0.523 | 0.772 | 0.925 | 0.914 | 0.691 | 0.837 |
| Weather | 96 | 0.151 | 0.201 | 0.154 | 0.205 | **0.146** | **0.194** | 0.151 | 0.202 | 0.216 | 0.280 | 0.292 | 0.370 | 0.153 | 0.211 | 0.215 | 0.296 | 0.327 | 0.359 | 0.433 | 0.462 |
| | 192 | 0.195 | 0.242 | 0.200 | 0.247 | **0.192** | **0.238** | 0.223 | 0.295 | 0.303 | 0.335 | 0.410 | 0.473 | 0.207 | 0.250 | 0.267 | 0.345 | 0.390 | 0.422 | 0.508 | 0.518 |
| | 336 | 0.244 | **0.280** | 0.245 | 0.282 | **0.245** | **0.280** | 0.246 | 0.283 | 0.351 | 0.358 | 0.434 | 0.427 | 0.249 | 0.264 | 0.299 | 0.360 | 0.477 | 0.446 | 0.545 | 0.549 |
| | 720 | 0.314 | 0.330 | 0.312 | **0.330** | 0.320 | 0.336 | 0.320 | 0.338 | 0.425 | 0.399 | 0.539 | 0.523 | 0.319 | 0.320 | 0.361 | 0.395 | 0.551 | 0.586 | 0.576 | 0.572 |
| Traffic | 96 | 0.372 | **0.258** | 0.374 | 0.266 | 0.393 | 0.275 | **0.368** | 0.262 | 0.431 | 0.482 | 0.559 | 0.454 | 0.706 | 0.385 | 0.613 | 0.340 | 0.751 | 0.431 | 0.321 | 0.367 |
| | 192 | 0.396 | 0.271 | 0.395 | 0.270 | 0.376 | 0.254 | 0.373 | 0.251 | 0.491 | 0.346 | 0.583 | 0.493 | 0.709 | 0.388 | 0.619 | 0.516 | 0.751 | 0.424 | 0.476 | 0.367 |
| | 336 | 0.411 | 0.280 | 0.408 | 0.277 | 0.384 | 0.259 | **0.395** | **0.254** | 0.502 | 0.384 | 0.637 | 0.469 | 0.714 | 0.394 | 0.785 | 0.497 | 0.761 | 0.425 | 0.499 | 0.376 |
| | 720 | 0.436 | **0.290** | 0.436 | **0.290** | 0.446 | 0.306 | **0.432** | **0.290** | 0.533 | 0.543 | 0.663 | 0.594 | 0.723 | 0.421 | 0.850 | 0.472 | 0.780 | 0.433 | 0.563 | 0.390 |
| Electricity | 96 | 0.130 | 0.225 | 0.131 | 0.226 | **0.126** | **0.221** | 0.133 | 0.223 | 0.399 | 0.412 | 0.292 | 0.370 | 0.166 | 0.254 | 0.366 | 0.436 | 0.230 | 0.353 | 0.322 | 0.401 |
| | 192 | **0.144** | 0.240 | 0.145 | 0.240 | 0.145 | 0.238 | 0.147 | **0.237** | 0.400 | 0.460 | 0.270 | 0.373 | 0.178 | 0.278 | 0.366 | 0.433 | 0.253 | 0.371 | 0.343 | 0.416 |
| | 336 | **0.160** | **0.256** | 0.162 | 0.256 | 0.164 | 0.256 | 0.166 | 0.265 | 0.564 | 0.573 | 0.334 | 0.323 | 0.186 | 0.275 | 0.358 | 0.428 | 0.197 | 0.287 | 0.362 | 0.435 |
| | 720 | **0.194** | **0.287** | 0.201 | 0.290 | 0.200 | 0.290 | 0.203 | 0.297 | 0.880 | 0.770 | 0.344 | 0.346 | 0.213 | 0.288 | 0.363 | 0.431 | 0.230 | 0.328 | 0.388 | 0.456 |
| Average | | **0.301** | **0.327** | 0.304 | 0.328 | 0.314 | 0.333 | 0.306 | 0.331 | 0.522 | 0.475 | 0.473 | 0.452 | 0.388 | 0.368 | 0.507 | 0.501 | 0.560 | 0.518 | 0.593 | 0.565 |

\* We used the official code to replicate the results.  † SimMTM is a concurrent work to ours.

Table D.1: **Results of multivariate TSF with self-supervised methods.** The best results are in bold and the second best are underlined.

# E  TRANSFER LEARNING

For time series forecasting under transfer learning, we consider both in-domain and cross-domain transfer learning settings, where we consider datasets with same frequency as in-domain. We perform transfer learning in both in-domain and cross-domain using five datasets: four ETT datasests and Weather. The full results are described in Table E.1, where missing values are not reported in literature.

| | | | PITS | | | | | | PatchTST | | | | | | SimMTM | | TimeMAE | | TST | | LaST | | TF-C | | CoST | | TS2Vec | |
| | | | FT | | LP | | SL | | FT | | LP | | SL | | | | | | | | | | | | | | | |
| source | target | horizon | MSE | MAE | MSE | MAE | MSE | MAE | MSE | MAE | MSE | MAE | MSE | MAE | MSE | MAE | MSE | MAE | MSE | MAE | MSE | MAE | MSE | MAE | MSE | MAE | MSE | MAE |
|---|---|---|---|---|---|---|---|---|---|---|---|---|---|---|---|---|---|---|---|---|---|---|---|---|---|---|---|---|
| In-domain | ETTh2→ETTh1 | 96 | 0.367 | 0.394 | 0.367 | 0.393 | 0.378 | 0.400 | 0.380 | 0.411 | 0.405 | 0.426 | 0.458 | 0.443 | 0.372 | 0.402 | 0.703 | 0.562 | 0.653 | 0.468 | 0.362 | 0.420 | 0.596 | 0.569 | 0.378 | 0.421 | 0.849 | 0.694 |
| | | 192 | 0.398 | 0.414 | 0.398 | 0.413 | 0.419 | 0.420 | 0.419 | 0.436 | 0.433 | 0.443 | 0.514 | 0.472 | 0.414 | 0.425 | 0.715 | 0.567 | 0.658 | 0.502 | 0.426 | 0.478 | 0.614 | 0.621 | 0.424 | 0.451 | 0.909 | 0.738 |
| | | 336 | 0.416 | 0.428 | 0.417 | 0.429 | 0.433 | 0.431 | 0.436 | 0.449 | 0.447 | 0.456 | 0.559 | 0.498 | 0.429 | 0.436 | 0.733 | 0.579 | 0.631 | 0.561 | 0.522 | 0.509 | 0.694 | 0.664 | 0.651 | 0.582 | 1.082 | 0.775 |
| | | 720 | 0.437 | 0.455 | 0.428 | 0.453 | 0.459 | 0.464 | 0.457 | 0.474 | 0.572 | 0.540 | 0.507 | 0.490 | 0.446 | 0.458 | 0.762 | 0.622 | 0.638 | 0.608 | 0.460 | 0.478 | 0.635 | 0.683 | 0.883 | 0.701 | 0.934 | 0.769 |
| | | avg | 0.404 | 0.423 | 0.403 | 0.422 | 0.422 | 0.429 | 0.423 | 0.443 | 0.464 | 0.466 | 0.510 | 0.476 | 0.415 | 0.430 | 0.728 | 0.583 | 0.645 | 0.535 | 0.443 | 0.471 | 0.635 | 0.634 | 0.584 | 0.539 | 0.944 | 0.744 |
| | ETTm2→ETTm1 | 96 | 0.305 | 0.358 | 0.308 | 0.350 | 0.302 | 0.352 | 0.294 | 0.350 | 0.294 | 0.350 | 0.327 | 0.360 | 0.297 | 0.348 | 0.647 | 0.497 | 0.471 | 0.422 | 0.304 | 0.388 | 0.610 | 0.577 | 0.239 | 0.331 | 0.586 | 0.515 |
| | | 192 | 0.331 | 0.379 | 0.338 | 0.369 | 0.342 | 0.377 | 0.333 | 0.371 | 0.330 | 0.372 | 0.393 | 0.398 | 0.332 | 0.370 | 0.597 | 0.508 | 0.495 | 0.442 | 0.429 | 0.494 | 0.725 | 0.657 | 0.339 | 0.371 | 0.624 | 0.562 |
| | | 336 | 0.363 | 0.388 | 0.364 | 0.385 | 0.373 | 0.391 | 0.359 | 0.392 | 0.359 | 0.386 | 0.425 | 0.425 | 0.364 | 0.393 | 0.700 | 0.525 | 0.455 | 0.424 | 0.499 | 0.523 | 0.768 | 0.684 | 0.371 | 0.421 | 1.035 | 0.806 |
| | | 720 | 0.401 | 0.409 | 0.405 | 0.408 | 0.422 | 0.420 | 0.407 | 0.414 | 0.406 | 0.415 | 0.500 | 0.473 | 0.410 | 0.431 | 0.786 | 0.596 | 0.498 | 0.532 | 0.422 | 0.450 | 0.927 | 0.759 | 0.467 | 0.481 | 0.780 | 0.669 |
| | | avg | 0.345 | 0.378 | 0.354 | 0.379 | 0.359 | 0.386 | 0.348 | 0.382 | 0.347 | 0.381 | 0.411 | 0.414 | 0.351 | 0.383 | 0.682 | 0.531 | 0.480 | 0.455 | 0.414 | 0.464 | 0.758 | 0.669 | 0.354 | 0.401 | 0.756 | 0.638 |
| Cross-domain | ETTm2→ETTh1 | 96 | 0.371 | 0.399 | 0.369 | 0.397 | 0.381 | 0.405 | 0.385 | 0.411 | 0.379 | 0.408 | 0.450 | 0.436 | 0.388 | 0.421 | 0.699 | 0.566 | 0.559 | 0.489 | 0.428 | 0.454 | 0.968 | 0.738 | 0.377 | 0.419 | 0.783 | 0.669 |
| | | 192 | 0.405 | 0.423 | 0.402 | 0.419 | 0.417 | 0.429 | 0.425 | 0.439 | 0.414 | 0.430 | 0.504 | 0.466 | 0.419 | 0.423 | 0.722 | 0.573 | 0.600 | 0.579 | 0.427 | 0.497 | 1.080 | 0.801 | 0.422 | 0.450 | 0.828 | 0.691 |
| | | 336 | 0.417 | 0.442 | 0.416 | 0.441 | 0.439 | 0.444 | 0.440 | 0.451 | 0.431 | 0.446 | 0.543 | 0.483 | 0.435 | 0.444 | 0.714 | 0.569 | 0.677 | 0.572 | 0.528 | 0.540 | 1.091 | 0.824 | 0.648 | 0.580 | 0.990 | 0.762 |
| | | 720 | 0.433 | 0.460 | 0.433 | 0.461 | 0.480 | 0.488 | 0.482 | 0.488 | 0.460 | 0.476 | 0.523 | 0.502 | 0.468 | 0.474 | 0.760 | 0.611 | 0.694 | 0.664 | 0.527 | 0.537 | 1.226 | 0.893 | 0.880 | 0.699 | 0.985 | 0.783 |
| | | avg | 0.407 | 0.431 | 0.405 | 0.430 | 0.429 | 0.441 | 0.433 | 0.447 | 0.421 | 0.440 | 0.505 | 0.472 | 0.428 | 0.441 | 0.724 | 0.580 | 0.632 | 0.576 | 0.503 | 0.507 | 1.091 | 0.814 | 0.582 | 0.537 | 0.896 | 0.726 |
| | ETTh2→ETTm1 | 96 | 0.300 | 0.354 | 0.304 | 0.346 | 0.294 | 0.347 | 0.302 | 0.353 | 0.326 | 0.372 | 0.326 | 0.361 | 0.322 | 0.347 | 0.658 | 0.505 | 0.449 | 0.343 | 0.314 | 0.396 | 0.677 | 0.603 | 0.253 | 0.342 | 0.466 | 0.480 |
| | | 192 | 0.335 | 0.374 | 0.335 | 0.364 | 0.332 | 0.367 | 0.342 | 0.375 | 0.354 | 0.386 | 0.371 | 0.392 | 0.332 | 0.375 | 0.594 | 0.511 | 0.477 | 0.407 | 0.587 | 0.545 | 0.718 | 0.638 | 0.367 | 0.392 | 0.557 | 0.532 |
| | | 336 | 0.361 | 0.393 | 0.367 | 0.383 | 0.363 | 0.387 | 0.370 | 0.392 | 0.392 | 0.409 | 0.413 | 0.418 | 0.394 | 0.391 | 0.732 | 0.532 | 0.407 | 0.519 | 0.631 | 0.584 | 0.755 | 0.663 | 0.388 | 0.431 | 0.646 | 0.576 |
| | | 720 | 0.404 | 0.417 | 0.423 | 0.414 | 0.363 | 0.419 | 0.439 | 0.426 | 0.440 | 0.434 | 0.486 | 0.460 | 0.411 | 0.424 | 0.768 | 0.592 | 0.557 | 0.523 | 0.468 | 0.429 | 0.848 | 0.712 | 0.498 | 0.488 | 0.752 | 0.638 |
| | | avg | 0.350 | 0.384 | 0.357 | 0.377 | 0.352 | 0.380 | 0.363 | 0.387 | 0.378 | 0.400 | 0.399 | 0.408 | 0.365 | 0.384 | 0.688 | 0.535 | 0.472 | 0.448 | 0.475 | 0.489 | 0.750 | 0.654 | 0.377 | 0.413 | 0.606 | 0.556 |
| | ETTm1→ETTh1 | 96 | 0.382 | 0.402 | 0.375 | 0.398 | 0.382 | 0.402 | 0.388 | 0.411 | 0.373 | 0.401 | 0.456 | 0.442 | 0.367 | 0.398 | 0.715 | 0.581 | 0.627 | 0.477 | 0.360 | 0.374 | 0.666 | 0.647 | 0.423 | 0.450 | 0.991 | 0.765 |
| | | 192 | 0.405 | 0.423 | 0.409 | 0.420 | 0.417 | 0.421 | 0.422 | 0.431 | 0.408 | 0.423 | 0.520 | 0.482 | 0.396 | 0.421 | 0.729 | 0.587 | 0.628 | 0.500 | 0.381 | 0.371 | 0.672 | 0.653 | 0.641 | 0.578 | 0.829 | 0.699 |
| | | 336 | 0.415 | 0.435 | 0.423 | 0.442 | 0.441 | 0.436 | 0.449 | 0.449 | 0.448 | 0.452 | 0.544 | 0.494 | 0.471 | 0.437 | 0.712 | 0.583 | 0.683 | 0.554 | 0.472 | 0.531 | 0.626 | 0.711 | 0.863 | 0.694 | 0.971 | 0.787 |
| | | 720 | 0.433 | 0.463 | 0.428 | 0.459 | 0.451 | 0.461 | 0.530 | 0.513 | 0.499 | 0.492 | 0.532 | 0.507 | 0.454 | 0.463 | 0.747 | 0.627 | 0.642 | 0.600 | 0.490 | 0.488 | 0.835 | 0.797 | 1.071 | 0.805 | 1.037 | 0.820 |
| | | avg | 0.406 | 0.427 | 0.407 | 0.428 | 0.422 | 0.430 | 0.447 | 0.451 | 0.432 | 0.442 | 0.513 | 0.481 | 0.422 | 0.430 | 0.726 | 0.595 | 0.645 | 0.533 | 0.426 | 0.441 | 0.700 | 0.702 | 0.750 | 0.632 | 0.957 | 0.768 |
| | ETTh1→ETTm1 | 96 | 0.300 | 0.353 | 0.303 | 0.347 | 0.299 | 0.352 | 0.293 | 0.344 | 0.316 | 0.359 | 0.322 | 0.360 | 0.290 | 0.348 | 0.667 | 0.521 | 0.425 | 0.381 | 0.295 | 0.387 | 0.672 | 0.600 | 0.248 | 0.332 | 0.605 | 0.561 |
| | | 192 | 0.339 | 0.376 | 0.334 | 0.364 | 0.334 | 0.371 | 0.327 | 0.366 | 0.351 | 0.378 | 0.388 | 0.399 | 0.327 | 0.372 | 0.561 | 0.479 | 0.495 | 0.478 | 0.335 | 0.379 | 0.721 | 0.639 | 0.336 | 0.391 | 0.615 | 0.561 |
| | | 336 | 0.364 | 0.389 | 0.367 | 0.383 | 0.365 | 0.392 | 0.364 | 0.397 | 0.386 | 0.399 | 0.408 | 0.415 | 0.357 | 0.392 | 0.690 | 0.533 | 0.456 | 0.441 | 0.379 | 0.363 | 0.755 | 0.664 | 0.381 | 0.421 | 0.763 | 0.677 |
| | | 720 | 0.404 | 0.418 | 0.424 | 0.415 | 0.424 | 0.419 | 0.409 | 0.417 | 0.441 | 0.430 | 0.491 | 0.464 | 0.409 | 0.423 | 0.744 | 0.583 | 0.554 | 0.477 | 0.403 | 0.431 | 0.837 | 0.705 | 0.469 | 0.482 | 0.805 | 0.664 |
| | | avg | 0.353 | 0.384 | 0.357 | 0.377 | 0.356 | 0.384 | 0.348 | 0.381 | 0.374 | 0.392 | 0.402 | 0.410 | 0.346 | 0.384 | 0.666 | 0.529 | 0.482 | 0.444 | 0.353 | 0.390 | 0.746 | 0.652 | 0.359 | 0.407 | 0.697 | 0.616 |
| | Weather→ETTh1 | 96 | 0.373 | 0.398 | 0.373 | 0.398 | 0.379 | 0.401 | 0.386 | 0.409 | 0.384 | 0.401 | 0.469 | 0.444 | 0.477 | 0.444 | - | - | - | - | - | - | - | - | - | - | - | - |
| | | 192 | 0.410 | 0.420 | 0.407 | 0.420 | 0.408 | 0.419 | 0.405 | 0.420 | 0.408 | 0.422 | 0.518 | 0.476 | 0.454 | 0.452 | - | - | - | - | - | - | - | - | - | - | - | - |
| | | 336 | 0.415 | 0.433 | 0.415 | 0.433 | 0.421 | 0.436 | 0.448 | 0.454 | 0.421 | 0.438 | 0.551 | 0.497 | 0.424 | 0.434 | - | - | - | - | - | - | - | - | - | - | - | - |
| | | 720 | 0.428 | 0.457 | 0.428 | 0.457 | 0.477 | 0.480 | 0.508 | 0.508 | 0.479 | 0.489 | 0.542 | 0.507 | 0.468 | 0.469 | - | - | - | - | - | - | - | - | - | - | - | - |
| | | avg | 0.407 | 0.427 | 0.407 | 0.427 | 0.421 | 0.434 | 0.437 | 0.448 | 0.423 | 0.438 | 0.520 | 0.481 | 0.456 | 0.467 | - | - | - | - | - | - | - | - | - | - | - | - |
| | Weather→ETTm1 | 96 | 0.295 | 0.353 | 0.307 | 0.350 | 0.299 | 0.354 | 0.292 | 0.347 | 0.300 | 0.351 | 0.339 | 0.365 | 0.304 | 0.354 | - | - | - | - | - | - | - | - | - | - | - | - |
| | | 192 | 0.329 | 0.371 | 0.336 | 0.366 | 0.342 | 0.384 | 0.332 | 0.373 | 0.336 | 0.372 | 0.381 | 0.395 | 0.338 | 0.375 | - | - | - | - | - | - | - | - | - | - | - | - |
| | | 336 | 0.354 | 0.387 | 0.365 | 0.383 | 0.365 | 0.390 | 0.360 | 0.391 | 0.370 | 0.392 | 0.423 | 0.423 | 0.371 | 0.397 | - | - | - | - | - | - | - | - | - | - | - | - |
| | | 720 | 0.418 | 0.420 | 0.413 | 0.410 | 0.418 | 0.421 | 0.406 | 0.421 | 0.413 | 0.425 | 0.506 | 0.466 | 0.417 | 0.426 | - | - | - | - | - | - | - | - | - | - | - | - |
| | | avg | 0.351 | 0.386 | 0.356 | 0.378 | 0.356 | 0.388 | 0.348 | 0.383 | 0.355 | 0.385 | 0.412 | 0.412 | 0.358 | 0.388 | - | - | - | - | - | - | - | - | - | - | - | - |

Table E.1: **Results of multivariate TS forecasting with transfer learning.** We conduct experiments under two settings: (1) in-domain and (2) cross-domain transfer. The best results are in bold and the second best are underlined.

# F   COMPARISON WITH PATCHTST

We compare our proposed method with PatchTST in three versions: 1) fine-tuning (FT), linear probing (LP), and supervised learning (SL). The results are described in Table F.1, which demonstrates that our proposed method outperforms PatchTST in every version in most of the datasets.

| Models | Metric | PITS | | | | | | PatchTST | | | | | |
|---|---|---|---|---|---|---|---|---|---|---|---|---|---|
| | | FT | | LP | | SL | | FT | | LP | | SL | |
| | | MSE | MAE | MSE | MAE | MSE | MAE | MSE | MAE | MSE | MAE | MSE | MAE |
| ETTh1 | 96 | 0.367 | 0.393 | **0.366** | **0.392** | 0.369 | 0.397 | 0.379 | 0.408 | 0.382 | 0.410 | 0.375 | 0.399 |
| | 192 | 0.401 | 0.416 | **0.398** | **0.414** | 0.403 | 0.416 | 0.414 | 0.428 | 0.433 | 0.441 | 0.414 | 0.421 |
| | 336 | 0.415 | 0.428 | 0.419 | 0.427 | **0.409** | **0.426** | 0.435 | 0.446 | 0.439 | 0.446 | 0.431 | 0.436 |
| | 720 | **0.425** | **0.452** | 0.430 | 0.454 | 0.456 | 0.465 | 0.468 | 0.474 | 0.482 | 0.482 | 0.449 | 0.466 |
| | avg | **0.401** | **0.421** | 0.403 | 0.422 | 0.409 | 0.426 | 0.424 | 0.439 | 0.434 | 0.445 | 0.417 | 0.431 |
| ETTh2 | 96 | **0.269** | **0.333** | **0.269** | **0.333** | 0.281 | 0.343 | 0.306 | 0.351 | 0.299 | 0.350 | 0.274 | 0.336 |
| | 192 | **0.329** | **0.371** | 0.331 | 0.373 | 0.345 | 0.383 | 0.361 | 0.392 | 0.363 | 0.394 | 0.339 | 0.379 |
| | 336 | 0.356 | 0.397 | 0.352 | 0.395 | 0.334 | 0.389 | 0.405 | 0.427 | 0.386 | 0.417 | 0.331 | 0.380 |
| | 720 | 0.383 | 0.425 | 0.383 | 0.425 | 0.389 | 0.430 | 0.419 | 0.446 | 0.409 | 0.440 | 0.379 | 0.422 |
| | avg | **0.334** | 0.382 | 0.335 | 0.381 | 0.337 | 0.386 | 0.373 | 0.404 | 0.364 | 0.400 | 0.331 | 0.379 |
| ETTm1 | 96 | 0.294 | 0.354 | 0.307 | 0.349 | 0.296 | 0.346 | 0.294 | 0.345 | 0.296 | 0.349 | 0.290 | 0.342 |
| | 192 | 0.321 | 0.373 | 0.337 | 0.368 | 0.330 | 0.369 | 0.327 | 0.369 | 0.333 | 0.370 | 0.332 | 0.369 |
| | 336 | 0.359 | 0.388 | 0.365 | 0.389 | 0.360 | 0.388 | 0.364 | 0.390 | 0.368 | 0.390 | 0.366 | 0.392 |
| | 720 | 0.396 | 0.414 | 0.415 | 0.412 | 0.416 | 0.421 | 0.409 | 0.415 | 0.422 | 0.418 | 0.420 | 0.424 |
| | avg | 0.342 | 0.381 | 0.356 | 0.379 | 0.351 | 0.381 | 0.349 | 0.380 | 0.355 | 0.382 | 0.352 | 0.382 |
| ETTm2 | 96 | 0.165 | 0.260 | 0.160 | 0.252 | 0.163 | 0.255 | 0.167 | 0.256 | 0.168 | 0.257 | 0.165 | 0.255 |
| | 192 | 0.213 | 0.291 | 0.214 | 0.289 | 0.215 | 0.293 | 0.232 | 0.302 | 0.231 | 0.302 | 0.220 | 0.292 |
| | 336 | 0.263 | 0.325 | 0.263 | 0.324 | 0.266 | 0.329 | 0.291 | 0.342 | 0.290 | 0.341 | 0.278 | 0.329 |
| | 720 | 0.337 | 0.373 | 0.342 | 0.376 | 0.342 | 0.380 | 0.368 | 0.390 | 0.366 | 0.387 | 0.367 | 0.385 |
| | avg | 0.244 | 0.310 | 0.244 | 0.310 | 0.247 | 0.314 | 0.264 | 0.323 | 0.264 | 0.322 | 0.258 | 0.315 |
| Weather | 96 | 0.151 | 0.201 | 0.167 | 0.222 | 0.154 | 0.202 | 0.146 | 0.194 | 0.160 | 0.211 | 0.152 | 0.199 |
| | 192 | 0.195 | 0.242 | 0.211 | 0.259 | 0.191 | 0.242 | 0.192 | 0.238 | 0.203 | 0.248 | 0.197 | 0.243 |
| | 336 | 0.244 | 0.280 | 0.256 | 0.293 | 0.245 | 0.280 | 0.245 | 0.280 | 0.251 | 0.285 | 0.249 | 0.283 |
| | 720 | 0.314 | 0.330 | 0.319 | 0.338 | 0.309 | 0.330 | 0.320 | 0.336 | 0.319 | 0.334 | 0.320 | 0.335 |
| | avg | 0.225 | 0.262 | 0.239 | 0.278 | 0.225 | 0.263 | 0.226 | 0.262 | 0.233 | 0.269 | 0.230 | 0.265 |
| Traffic | 96 | 0.372 | 0.258 | 0.384 | 0.266 | 0.375 | 0.264 | 0.393 | 0.275 | 0.399 | 0.294 | 0.367 | 0.251 |
| | 192 | 0.396 | 0.271 | 0.395 | 0.270 | 0.389 | 0.270 | 0.376 | 0.254 | 0.412 | 0.298 | 0.385 | 0.259 |
| | 336 | 0.411 | 0.280 | 0.409 | 0.276 | 0.401 | 0.277 | 0.384 | 0.259 | 0.425 | 0.306 | 0.398 | 0.265 |
| | 720 | 0.436 | 0.290 | 0.438 | 0.295 | 0.437 | 0.294 | 0.446 | 0.306 | 0.460 | 0.323 | 0.434 | 0.287 |
| | avg | 0.403 | 0.271 | 0.406 | 0.267 | 0.401 | 0.276 | 0.400 | 0.274 | 0.424 | 0.305 | 0.396 | 0.266 |
| Electricity | 96 | 0.130 | 0.225 | 0.132 | 0.228 | 0.132 | 0.228 | 0.126 | 0.221 | 0.138 | 0.237 | 0.130 | 0.222 |
| | 192 | 0.144 | 0.240 | 0.147 | 0.242 | 0.147 | 0.242 | 0.145 | 0.238 | 0.156 | 0.252 | 0.148 | 0.240 |
| | 336 | 0.160 | 0.256 | 0.163 | 0.258 | 0.162 | 0.261 | 0.164 | 0.256 | 0.170 | 0.265 | 0.167 | 0.261 |
| | 720 | 0.197 | 0.290 | 0.201 | 0.290 | 0.199 | 0.290 | 0.200 | 0.290 | 0.208 | 0.297 | 0.202 | 0.291 |
| | avg | 0.157 | 0.253 | 0.161 | 0.254 | 0.160 | 0.256 | 0.159 | 0.251 | 0.168 | 0.263 | 0.162 | 0.254 |
| Average | | 0.301 | 0.327 | 0.306 | 0.328 | 0.304 | 0.329 | 0.314 | 0.333 | 0.320 | 0.341 | 0.307 | 0.327 |

Table F.1: PITS vs. PatchTST in multivariate time series forecasting.

# G EFFECTIVENESS OF PI TASK AND CONTRASTIVE LEARNING

To assess the effectiveness of the proposed patch reconstruction task and complementary contrastive learning, we conduct ablation studies in both time series forecasting and time series classification.

## G.1 TIME SERIES FORECASTING

To examine the effect of PI task and CL on forecasting, we conduct an experiment using four ETT datasets. The results in Table G.1 demonstrate that performing CL with the representation obtained from the first layer and PI with the one from the second layer gives the best performance.

| Layer 1 | - | - | - | PI | CL |
|---|---|---|---|---|---|
| Layer 2 | CL | PI | CL+PI | CL | PI |
| ETTh1 96 | 0.715 | **0.367** | 0.372 | 0.381 | **0.367** |
| ETTh1 192 | 0.720 | **0.400** | 0.409 | 0.416 | 0.401 |
| ETTh1 336 | 0.719 | 0.426 | 0.422 | 0.462 | **0.415** |
| ETTh1 720 | 0.727 | 0.443 | 0.465 | 0.509 | **0.425** |
| ETTh1 avg | 0.720 | 0.409 | 0.417 | 0.442 | **0.401** |
| ETTh2 96 | 0.373 | 0.270 | 0.307 | 0.303 | **0.269** |
| ETTh2 192 | 0.384 | 0.331 | 0.362 | 0.373 | **0.329** |
| ETTh2 336 | 0.386 | 0.361 | 0.387 | 0.391 | **0.356** |
| ETTh2 720 | 0.432 | 0.384 | 0.408 | 0.416 | **0.383** |
| ETTh2 avg | 0.394 | 0.336 | 0.366 | 0.371 | **0.334** |
| ETTm1 96 | 0.693 | 0.305 | 0.302 | 0.300 | **0.294** |
| ETTm1 192 | 0.702 | 0.335 | 0.337 | 0.336 | **0.321** |
| ETTm1 336 | 0.716 | 0.366 | 0.365 | 0.369 | **0.359** |
| ETTm1 720 | 0.731 | 0.413 | 0.413 | 0.426 | **0.396** |
| ETTm1 avg | 0.711 | 0.355 | 0.356 | 0.358 | **0.342** |
| ETTm2 96 | 0.346 | **0.160** | 0.167 | 0.171 | 0.165 |
| ETTm2 192 | 0.368 | 0.215 | 0.225 | 0.235 | **0.213** |
| ETTm2 336 | 0.397 | 0.266 | 0.274 | 0.278 | **0.263** |
| ETTm2 720 | 0.424 | 0.346 | 0.351 | 0.376 | **0.337** |
| ETTm2 avg | 0.381 | 0.247 | 0.254 | 0.265 | **0.244** |
| Total avg | 0.552 | 0.337 | 0.348 | 0.359 | **0.330** |

Table G.1: Effect of PI task and CL on time series forecasting.

## G.2 TIME SERIES CLASSIFICATION

To evaluate the impact of employing CL and PI on classification, we conducted an experiment using the Epilepsy dataset. The results presented in Table G.2 demonstrate that as long as PI task is employed, the performance is robust to the design choices.

| Layer 1 | - | - | - | PI | CL |
|---|---|---|---|---|---|
| Layer 2 | CL | PI | CL+PI | CL | PI |
| SleepEEG ACC. | 91.61 | 95.27 | **95.67** | **95.67** | **95.67** |
| SleepEEG PRE. | 92.11 | 95.35 | 95.63 | **95.70** | 95.63 |
| SleepEEG REC.. | 91.61 | 95.27 | 95.66 | 95.66 | **95.67** |
| SleepEEG F1.. | 91.79 | 95.30 | **95.68** | **95.68** | 95.64 |

Table G.2: Effect of PI task and CL on time series classification.

## H    EFFECTIVENESS OF PI STRATEGIES

In this experiment, we investigate the impact of our proposed PI strategies from two perspectives: 1) the pretraining task and 2) the encoder architecture. The results, shown in Table H.1, encompass four ETT datasets with four different forecasting horizons with a common input horizon of 512. These results demonstrate that the PI task consistently outperforms the conventional PD task across all considered architectures.

| Architecture | | PI | | | PD | | | |
| --- | --- | --- | --- | --- | --- | --- | --- | --- |
| | | Linear | | MLP | | MLPMixer | | Transformer | |
| Task | | PD | PI | PD | PI | PD | PI | PD | PI |
| ETTh1 | 96 | 0.366 | 0.365 | 0.375 | 0.366 | 0.378 | 0.368 | 0.371 | 0.372 |
| | 192 | 0.398 | 0.398 | 0.407 | 0.397 | 0.414 | 0.399 | 0.410 | 0.404 |
| | 336 | 0.423 | 0.424 | 0.427 | 0.427 | 0.422 | 0.427 | 0.443 | 0.434 |
| | 720 | 0.444 | 0.444 | 0.463 | 0.440 | 0.465 | 0.440 | 0.475 | 0.452 |
| | avg | **0.408** | **0.408** | 0.418 | **0.407** | 0.420 | **0.409** | 0.425 | **0.415** |
| ETTh2 | 96 | 0.272 | 0.270 | 0.290 | 0.270 | 0.301 | 0.276 | 0.283 | 0.271 |
| | 192 | 0.332 | 0.333 | 0.361 | 0.329 | 0.353 | 0.334 | 0.351 | 0.332 |
| | 336 | 0.370 | 0.364 | 0.373 | 0.353 | 0.394 | 0.363 | 0.378 | 0.369 |
| | 720 | 0.396 | 0.385 | 0.418 | 0.384 | 0.411 | 0.389 | 0.400 | 0.395 |
| | avg | 0.343 | **0.338** | 0.361 | **0.334** | 0.365 | **0.341** | 0.353 | **0.342** |
| ETTm1 | 96 | 0.304 | 0.304 | 0.298 | 0.302 | 0.294 | 0.296 | 0.294 | 0.297 |
| | 192 | 0.337 | 0.338 | 0.341 | 0.337 | 0.332 | 0.334 | 0.335 | 0.336 |
| | 336 | 0.370 | 0.368 | 0.368 | 0.363 | 0.364 | 0.363 | 0.365 | 0.359 |
| | 720 | 0.423 | 0.423 | 0.416 | 0.420 | 0.418 | 0.416 | 0.405 | 0.403 |
| | avg | 0.359 | **0.358** | 0.356 | **0.355** | 0.354 | **0.352** | **0.350** | **0.350** |
| ETTm2 | 96 | 0.163 | 0.163 | 0.169 | 0.164 | 0.170 | 0.164 | 0.172 | 0.172 |
| | 192 | 0.219 | 0.218 | 0.224 | 0.218 | 0.226 | 0.218 | 0.240 | 0.221 |
| | 336 | 0.272 | 0.271 | 0.275 | 0.271 | 0.276 | 0.272 | 0.300 | 0.274 |
| | 720 | 0.362 | 0.361 | 0.363 | 0.359 | 0.361 | 0.359 | 0.383 | 0.356 |
| | avg | 0.254 | **0.253** | 0.258 | **0.253** | 0.259 | **0.253** | 0.274 | **0.256** |
| Total avg | | 0.341 | **0.339** | 0.348 | **0.337** | 0.350 | **0.339** | 0.351 | **0.341** |

Table H.1: Effectiveness of PI tasks and PI architectures.

## I    ROBUSTNESS TO PATCH SIZE

To evaluate the robustness of encoder architectures to patch size, we compare MLP and Transformer with different patch sizes with ETTh2 and ETTm2 with a common input horizon of 512. The left and the right panel of Figure I.1 illustrate the average MSE of four horizons of ETTh2 and ETTm2, respectively.

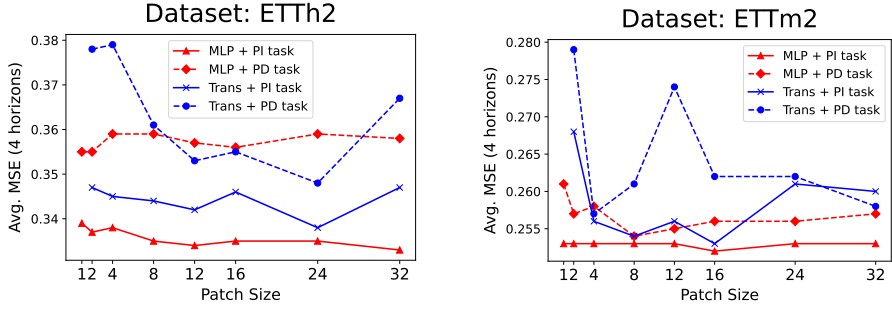

Figure I.1: Robustness of PI task to patch size.

## J  EFFICIENCY OF PITS IN SELF-SUPERVISED AND SUPERVISED SETTINGS

We compare the efficiency of PITS between self-supervised and supervised settings on the ETTm2 dataset. We calculate the pretraining time and fine-tuning time of PITS under the self-supervised setting, as well as the training time under the supervised setting. Table J.1 presents the results, with the time required for fine-tuning (in the self-supervised setting) and supervised training across four different horizons {96, 192, 336, 720}. We used an epoch size of 10 for both pretraining in self-supervised settings and training in supervised settings. For fine-tuning, we trained linear head for 10 epochs, followed by end-to-end fine-tuning of the entire network for an additional 20 epochs, following PatchTST. For self-supervised learning, we utilize a shared pretrained weight for all prediction horizons, enhancing efficiency over the long-term setting compared to supervised learning. Given that pretraining is done before training on downstream tasks, fine-tuning the pretrained model is more efficient than training from scratch, while providing better performance.

| | PITS | | | | | | | | |
| --- | --- | --- | --- | --- | --- | --- | --- | --- | --- |
| | Self-supervised (w/ hier. CL) | | | | | Supervised | | | |
| | Pretrain | Fine-tune | | | | Train | | | |
| Horizon | - | 96 | 192 | 336 | 720 | 96 | 192 | 336 | 720 |
| Time (min) | 16 | 1.2 | 1.4 | 1.5 | 1.6 | 4.2 | 4.3 | 5.3 | 6.9 |
| Avg. MSE | - | 0.244 | | | | 0.255 | | | |

Table J.1: Comparison of training time under self-supervised and supervised settings.

## K  PERFORMANCE BY DROPOUT RATE

Figure K.1 displays the average MSE across four horizons, and Table K.1 lists all the MSE values for four ETT datasets trained with MLP of $D = 32$ at various dropout rates with a common input horizon of 512. These results emphasize the importance of incorporating dropout during the pretraining phase of the reconstruction task, as it helps prevent trivial solutions when the hidden dimension is greater than the input dimension.

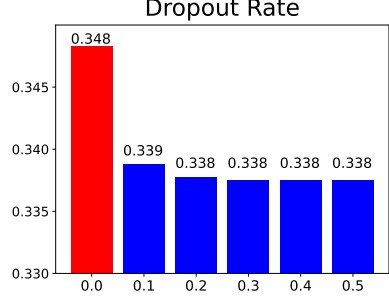

Figure K.1: Avg. MSE by dropout.

| Dropout rate | ETTh1 | ETTh2 | ETTm1 | ETTm2 | Avg. |
| --- | --- | --- | --- | --- | --- |
| 0.0 | 0.418 | 0.359 | 0.359 | 0.257 | 0.348 |
| 0.1 | 0.410 | 0.334 | 0.358 | 0.253 | 0.339 |
| 0.2 | 0.407 | 0.334 | 0.357 | 0.253 | 0.338 |
| 0.3 | 0.407 | 0.333 | 0.357 | 0.253 | 0.338 |
| 0.4 | 0.407 | 0.334 | 0.356 | 0.253 | 0.338 |
| 0.5 | 0.406 | 0.335 | 0.356 | 0.253 | **0.337** |

Table K.1: MSE by dropout.

## L  PERFORMANCE OF VARIOUS PRETRAIN TASKS

To see if the conventional PD task of reconstructing the masked patches ($X_m$) with the unmasked patches ($X_u$) is appropriate for TS representation learning, we employ two other simple pretraining tasks of 1) predicting $X_u$ with zero-value patches (**0**) and 2) reconstructing **0** with themselves. Table L.1 presents the results for four ETT datasets with a common input horizon of 512 across three different architectures: Transformer, MLP without CL, and MLP with CL. These results underscore that models pretarined with PD task performs even worse than the two basic pretraining tasks with zero-value patch inputs, highlighting the ineffectiveness of the PI task and emphasizing the importance of the proposed PI task.

| Pretrain Task | | Transformer | | | | | MLP | | | | | | | | | |
| | | | | | | | w/o CL | | | | | w/ CL | | | | |
| Input | Output | ETTh1 | ETTh2 | ETTm1 | ETTm2 | avg | ETTh1 | ETTh2 | ETTm1 | ETTm2 | avg | ETTh1 | ETTh2 | ETTm1 | ETTm2 | avg |
| $X_u$ | $X_u$ | 0.415 | 0.342 | 0.350 | 0.256 | **0.341** | 0.407 | 0.334 | 0.355 | 0.253 | **0.337** | 0.401 | 0.331 | 0.341 | 0.244 | **0.329** |
| $X_u$ | $X_m$ | 0.425 | 0.353 | 0.350 | 0.274 | 0.351 | 0.418 | 0.361 | 0.356 | 0.258 | 0.348 | 0.457 | 0.376 | 0.362 | 0.261 | 0.364 |
| **0** | $X_u$ | 0.410 | 0.350 | 0.349 | 0.260 | 0.342 | 0.418 | 0.361 | 0.354 | 0.256 | 0.348 | 0.418 | 0.361 | 0.353 | 0.256 | 0.348 |
| **0** | **0** | 0.413 | 0.360 | 0.342 | 0.257 | 0.343 | 0.418 | 0.356 | 0.352 | 0.253 | 0.345 | 0.418 | 0.356 | 0.353 | 0.254 | 0.345 |

Table L.1: Performance of various pretraining tasks.

## M  STATISTICS OF RESULTS OVER MULTIPLE RUNS

To see if the performance of PITS is consistent, we show the statistics of results with five different random seeds. We compute the mean and standard deviation of both MSE and MAE, as shown in Table M.1. The results indicate that the performance of PITS is consistent for both under self-supervised and supervised settings.

| Models | | Self-supervised | | Supervised | |
| | Metric | MSE | MAE | MSE | MAE |
| --- | --- | --- | --- | --- | --- |
| ETTh1 | 96 | $0.367_{\pm0.0035}$ | $0.393_{\pm0.0022}$ | $0.369_{\pm0.0011}$ | $0.397_{\pm0.0017}$ |
| | 192 | $0.401_{\pm0.0005}$ | $0.416_{\pm0.0008}$ | $0.403_{\pm0.0015}$ | $0.416_{\pm0.0020}$ |
| | 336 | $0.415_{\pm0.0021}$ | $0.428_{\pm0.0010}$ | $0.409_{\pm0.0002}$ | $0.426_{\pm0.0061}$ |
| | 720 | $0.425_{\pm0.0077}$ | $0.452_{\pm0.0045}$ | $0.456_{\pm0.0010}$ | $0.465_{\pm0.0022}$ |
| ETTh2 | 96 | $0.269_{\pm0.0013}$ | $0.333_{\pm0.0004}$ | $0.281_{\pm0.0009}$ | $0.343_{\pm0.0033}$ |
| | 192 | $0.329_{\pm0.0007}$ | $0.371_{\pm0.0015}$ | $0.345_{\pm0.0010}$ | $0.383_{\pm0.0040}$ |
| | 336 | $0.356_{\pm0.0021}$ | $0.397_{\pm0.0010}$ | $0.334_{\pm0.0019}$ | $0.389_{\pm0.0017}$ |
| | 720 | $0.383_{\pm0.0016}$ | $0.425_{\pm0.0005}$ | $0.389_{\pm0.0038}$ | $0.430_{\pm0.0025}$ |
| ETTm1 | 96 | $0.294_{\pm0.0027}$ | $0.354_{\pm0.0005}$ | $0.296_{\pm0.0011}$ | $0.346_{\pm0.0007}$ |
| | 192 | $0.321_{\pm0.0091}$ | $0.373_{\pm0.0035}$ | $0.330_{\pm0.0009}$ | $0.369_{\pm0.0010}$ |
| | 336 | $0.359_{\pm0.0029}$ | $0.383_{\pm0.0017}$ | $0.360_{\pm0.0005}$ | $0.388_{\pm0.0004}$ |
| | 720 | $0.396_{\pm0.0081}$ | $0.414_{\pm0.0060}$ | $0.416_{\pm0.0009}$ | $0.421_{\pm0.0014}$ |
| ETTm2 | 96 | $0.165_{\pm0.0017}$ | $0.260_{\pm0.0013}$ | $0.163_{\pm0.0005}$ | $0.255_{\pm0.0004}$ |
| | 192 | $0.213_{\pm0.0009}$ | $0.291_{\pm0.0011}$ | $0.215_{\pm0.0005}$ | $0.293_{\pm0.0004}$ |
| | 336 | $0.263_{\pm0.0002}$ | $0.325_{\pm0.0002}$ | $0.266_{\pm0.0002}$ | $0.329_{\pm0.0013}$ |
| | 720 | $0.337_{\pm0.0015}$ | $0.373_{\pm0.0003}$ | $0.342_{\pm0.0002}$ | $0.380_{\pm0.0015}$ |
| Weather | 96 | $0.151_{\pm0.0015}$ | $0.201_{\pm0.0027}$ | $0.154_{\pm0.0017}$ | $0.202_{\pm0.0005}$ |
| | 192 | $0.195_{\pm0.0011}$ | $0.242_{\pm0.0009}$ | $0.191_{\pm0.0015}$ | $0.242_{\pm0.0004}$ |
| | 336 | $0.244_{\pm0.0017}$ | $0.280_{\pm0.0017}$ | $0.245_{\pm0.0009}$ | $0.280_{\pm0.0004}$ |
| | 720 | $0.314_{\pm0.0016}$ | $0.330_{\pm0.0021}$ | $0.309_{\pm0.0010}$ | $0.330_{\pm0.0006}$ |
| Traffic | 96 | $0.372_{\pm0.0045}$ | $0.258_{\pm0.0033}$ | $0.375_{\pm0.0003}$ | $0.264_{\pm0.0002}$ |
| | 192 | $0.396_{\pm0.0001}$ | $0.271_{\pm0.0002}$ | $0.389_{\pm0.0002}$ | $0.270_{\pm0.0003}$ |
| | 336 | $0.411_{\pm0.0041}$ | $0.280_{\pm0.0030}$ | $0.401_{\pm0.0004}$ | $0.277_{\pm0.0001}$ |
| | 720 | $0.436_{\pm0.0061}$ | $0.290_{\pm0.0057}$ | $0.437_{\pm0.0003}$ | $0.294_{\pm0.0004}$ |
| Electricity | 96 | $0.130_{\pm0.0003}$ | $0.225_{\pm0.0003}$ | $0.132_{\pm0.0010}$ | $0.228_{\pm0.0011}$ |
| | 192 | $0.144_{\pm0.0008}$ | $0.240_{\pm0.0007}$ | $0.147_{\pm0.0008}$ | $0.242_{\pm0.0010}$ |
| | 336 | $0.160_{\pm0.0005}$ | $0.256_{\pm0.0006}$ | $0.162_{\pm0.0008}$ | $0.261_{\pm0.0019}$ |
| | 720 | $0.194_{\pm0.0003}$ | $0.287_{\pm0.0002}$ | $0.199_{\pm0.0006}$ | $0.290_{\pm0.0012}$ |

Table M.1: Results of PITS on multivariate TSF over five runs.

