# OpenReview forum: "Learning to Embed Time Series Patches Independently"
_ICLR.cc/2024/Conference — ICLR 2024 poster_

### Official Review · Reviewer_oFTu · 2023-10-25

**Soundness:** 2 fair
**Presentation:** 2 fair
**Contribution:** 2 fair
**Rating:** 5
**Confidence:** 2

**Summary:**

The key idea of the paper "Learning to Embed Time Series Patches Independently" is to propose a new approach for time series representation learning by focusing on embedding patches independently rather than capturing patch dependencies. The authors introduce a patch reconstruction task that autoencodes each patch without considering other patches and a patch-wise MLP architecture for embedding. They also incorporate complementary contrastive learning to efficiently capture adjacent time series information. Experimental results demonstrate that their proposed method outperforms state-of-the-art Transformer-based models in time series forecasting and classification tasks while being more efficient in terms of parameters and training time.

**Strengths:**

1) This paper is easy to follow.
2) The paper introduces the concept of patch independence for time series representation learning, arguing that learning to embed patches independently leads to better representations. This novel approach challenges the conventional strategy of capturing patch dependencies and proposes a patch reconstruction task and a patch-wise MLP architecture.
3) The paper presents extensive experiments on various tasks, including low-level and high-level tasks, demonstrating the superiority of the proposed method compared to state-of-the-art  models. The experiments are conducted under both standard and transfer learning settings, further validating the effectiveness of the approach (a significant performance gain is reported).

**Weaknesses:**

1) Despite the author's cautionary language, using only 'argue' rather than 'claim', 'learning to embed time series patches independently is superior to learning them dependently for TS representation learning' still sounds a bit lax to me. Especially considering the lack of substantial experimental evidence to support such an argument. (Comparison among representative methods cannot support the argument ‘A is superior to B when A,B refer to a set of methods’)

2) Given that you’ve mentioned the complementary masked strategy for CL and claim it as a ‘main contribution’, it should be investigated in related work. (Such strategy has already been proposed earlier, though in other domain. Such as Ref 1,2,3 provided below)

[1] SdAE: Self-distillated Masked Autoencoder

[2] Complementary Mask Self-Supervised Pre-training Based on Teacher-Student Network

[3] Deep Feature Selection Using a Novel Complementary Feature Mask

3) Lack of clarity in exposition (for certain sentences): The paper should strive to provide clear and concise explanations. For example, “Secondly, we propose to utilize the simple PI model architecture (e.g., MLP), as opposed to the conventional PD architecture (e.g., Transformer), which is not only more efficient but also performs better.”  What does which refer to in this sentence?

**Questions:**

Please refer to the weakness part. Further, in Table 2, does 46 mean 46 hour of training time?

---

> ### Author Response · Authors · 2023-11-18
>
> **W1. Support to the claim that PI is superior to PD for TS representation learning**
>
> Our observation is consistent across seven real-world datasets from various domains in the following four aspects, which is the main support to our claim (all table and figure indices below are based on the *new version* of our submission):
> - (1) Efficiency: **Table 13**, **Table I.1**
> - (2) Performance: **Table 2--8**
> - (3) Interpretability: **Figure 7**
> - (4) Robustness to distribution shift : **Figure 5**
> - (5) Robustness to patch size : **Figure 6**
>
> Nonetheless, we agree that this observation might not be extendable to some cases. We hope our proposed PI strategies provide a strong baseline for future TS representation learning works. For example, a future work could outperform ours by proposing more sophisticated PD strategies or benchmarks that require learning patch-dependence to perform well on downstream tasks. However, based on our empirical observation, it seems real-world datasets for TS analysis that we commonly use do not require learning patch-dependence.
>
> &nbsp;
>
> **W2. Related works on the complementary masked strategy**
>
> As noted by Reviewer oFTu, several complementary masked strategies have recently been proposed [A,B,C]. We initially omitted them as 1) our focus is primarily on the PI strategies, and 2) the role of the proposed CL is to boost the performance further as an auxiliary task, without hurting the PI task. However, we agree that discussion on those works is indeed interesting and helpful to remind audiences of recent progress, so we added a brief discussion on them in **Section 2**.
>
> SdAE [A] employs a student branch for information reconstruction and a teacher branch to generate latent representations of masked tokens, utilizing a complementary multi-fold masking strategy to maintain relevant mutual information between the branches. TSCAE [B] addresses the gap between upstream and downstream mismatches in the pretraining model based on MM by introducing complementary masks for teacher-student networks, and CFM [C] introduces a trainable complementary masking strategy for feature selection. However, our proposed complementary masking strategy differs in that it is not designed for a distillation model, and our masks are not learnable but randomly generated.
>
> [A] Chen, Yabo, et al. "Sdae: Self-distillated masked autoencoder." ECCV (2022).
>
> [B] Ye, Shaoxiong, Jing Huang, and Lifu Zhu. "Complementary Mask Self-Supervised Pre-training Based on Teacher-Student Network." ACCTCS (2023).
>
> [C] Liao, Yiwen, et al. "Deep Feature Selection Using a Novel Complementary Feature Mask." arXiv preprint arXiv:2209.12282 (2022).
>
> &nbsp;
>
> **W3. Clarity of sentences**
>
> We apologize for any confusion caused by certain sentences. For the example you provided "Secondly, we propose to utilize the simple PI model architecture (e.g., MLP), as opposed to the conventional PD architecture (e.g., Transformer), **which** is not only more efficient but also performs better.", **which** refers to **the simple PI model architecture**. We revised the sentence in the revision to clarify it. We will also keep improving the writing to make the final version clearer.
>
> &nbsp;
>
> **Q1.  Unit of training time**
>
> We apologize for the unclear expression. The unit of training time **Table 2 (of old version)** was sec/epoch, i.e., the time spent for one epoch of training in seconds. In the extended version of **Table 13 (of new version)**, we changed it to the total training time in minutes.

---

### Official Review · Reviewer_aEzH · 2023-10-29

**Soundness:** 3 good
**Presentation:** 3 good
**Contribution:** 3 good
**Rating:** 8
**Confidence:** 4

**Summary:**

The paper proposes a new approach called PITS (Patch Independence for Time Series) for self-supervised representation learning of time series data. The authors argue that previous methods that capture patch dependencies may not be optimal and instead propose learning to embed patches independently. They introduce the "Patch-Independence"-based MTM (Masked Time Series Modeling) method and complementary contrastive learning. They demonstrate that using an MLP as the PI architecture in MTM can independently reconstruct unmasked patches while disregarding interactions between patches, thereby improving efficiency while reducing the parameter count. Simultaneously, it has been shown that introducing complementary contrastive learning effectively captures information from adjacent time series. The model has been tested on two downstream tasks: time series forecasting and classification, demonstrating significant advantages in terms of performance, architectural interpretability, and robustness.

**Strengths:**

1) Originality: The paper introduces a novel approach to self-supervised time series representation learning, with a focus on patch independence. This represents a unique perspective challenging the traditional methods of capturing patch dependencies. Additionally, it is the first model to integrate MTM and CL, showcasing the innovation of the implementation while improving model performance.
2) Quality: The paper is well-written and provides a comprehensive analysis of the proposed method. The experimental design is thorough, including comparisons with state-of-the-art methods, and it demonstrates strong performance in two downstream tasks. Furthermore, the completeness of the ablation experiments illustrates attention to various model details.
3) Clarity: The paper is clear and well-organized. The authors provide a clear motivation for their work, explain the proposed method in detail, and present the experimental results in a clear and concise manner.
4) Significance: The proposed method achieves superior performance compared to existing approaches in time series forecasting and classification tasks. The efficiency of the method in terms of parameters and training time is also highlighted, which makes it more practical for real-world applications.

**Weaknesses:**

A substantive assessment of the weaknesses of the paper. Focus on constructive and actionable insights on how the work could improve towards its stated goals. Be specific, avoid generic remarks. For example, if you believe the contribution lacks novelty, provide references and an explanation as evidence: if you believe experiments are insufficient, explain why and exactly what is missing, etc.
1.	The representation in Figure 1(a) should be clearer and correspond one-to-one with the description in Section 3.2. For example, the inconsistency between "MLPmixer" in the figure and "MLP-Mixer" in the text should be rectified, and the explanations for both in the PI architecture need to be more explicit. In general, a more comprehensive explanation for this figure is needed, and citation markers may be introduced as necessary.
2.	The two layers described in Figure 1(b) should be clearly labeled, corresponding to one for the CL task and one for the PI task, as indicated in the caption.
3.	The notation "No(SL)" in Table 1 should be explained in the description, specifically in Section 2.
4.	One of the contributions is the use of self-supervised mask reconstruction, but for the TSF task, only two baseline models using self-supervised mask reconstruction are provided. The evaluation section could benefit from more comparisons with other baselines that use self-supervised mask reconstruction.

**Questions:**

1.	Could you provide a more detailed explanation for why the PI architecture is superior to the PD architecture? The results of the comparative experiments are quite significant, but we believe that adding some qualitative analysis would make the argument more convincing.
2.	Would it be possible to offer a more comprehensive explanation of how the proposed method is implemented, including details on hyperparameter configurations? We would appreciate seeing more comparative experimental results to further validate the rationale for these selections.
3.	Have you conducted comparisons between the proposed method and alternative baselines, such as autoencoders or other self-supervised learning approaches tailored for time series data?
4.	Have you taken into account the limitations of the proposed model? Could you elaborate on potential challenges or specific scenarios in which the method might exhibit suboptimal performance?

---

> ### Author Response · Authors · 2023-11-18
>
> **Q1. Explanation on why the PI architecture is superior to the PD architecture**
>
> We believe the PI arch is superior for a similar reason as pretraining with the PI task: **learning to embed time series patches independently is better.** The difference between the PI arch and PI task is in the stage where we force PI, i.e., by the arch design for the PI arch, or the learning objective for the PI task. Apart from the performance, another prominent advantage of the PI arch over PD arch is its efficiency, in terms of the number of model parameters and inference time shown in **Table 13 (of new version)**.
>
> As we correlated the advantage of PI arch and task above, here also we remind the qualitative intuition/analysis on why the PI task is superior:
> - 1. **Figure 1** provides us an intuition that pretraining with the PI task is more robust to distribution shifts.
> - 2. **Figure 5** provides us a more systematic analysis in terms of different trend and seasonality, where we observed that pretraining with the PD task becomes even worse when the slope is flipped and amplitude is increased.
> - 3. Though it is not about the performance, **Figure 7** shows that the PI arch is more interpretable than the PD arch.
>
> &nbsp;
>
> **Q2. Details of experimental settings**
>
> We added **Appendix B (of new version)** to provide the detail on experimental settings. We also clarify that we will release the code upon acceptance for reproducibility. However, if you find some hyperparameter or setting is still not clear, please feel free to share it.
>
> We also note that all hyperparameters are chosen by validation, where we added the information about validation in **Section 4.2**, and we provide ablation studies and analyses for our design choices through **Section 4.4 and 5**.
>
> &nbsp;
>
>
> **Q3. Comparison with self-supervised learning approaches other than MTM and CL**
>
> Unfortunately, we could not find other self-supervised learning approaches competitive to MTM and CL in recent works.
>
> We note that our ablation study in **Figure 4** includes **the comparison with the vanilla autoencoder, i.e., the one "w/o Dropout"**. However, following the conventional knowledge, autoencoders can learn good features when the size of the hidden dimension is small; otherwise, autoencoders might learn the trivial solution. **Figure 4** shows that the vanilla autoencoder is generally not better than ours as a pretext task.
>
> We agree that exploring or revisiting other self-supervised learning approaches would be an interesting future work.
>
>
> &nbsp;
>
>
> **Q4. Explanation of limitations and potential challenges**
>
> Thanks for bringing this to our attention. Though our extensive experiments demonstrate the effectiveness of the proposed PI strategies across multiple benchmark datasets that we have evaluated, this observation might not be extendable to some cases.
>
> For example, in our opinion, if a dataset has long-term dependencies, the dataset size is sufficiently large to learn such dependencies, and capturing the dependency is crucial for a downstream task, then a sophisticated PD strategy may outperform PI.
>
> In fact, the PD architecture can be seen as a generalization of the PI architecture, as learning to ignore patch-dependence completely in the PD architecture makes it operate as like PI. Hence, the PD architecture should be able to replicate the performance of the PI architecture given that other conditions are ideal (though we could not empirically find such ideal cases).
>
> We hope our proposed PI strategies provide a strong baseline for future TS representation learning works.

---

### Official Review · Reviewer_DsGC · 2023-11-01

**Soundness:** 3 good
**Presentation:** 3 good
**Contribution:** 3 good
**Rating:** 6
**Confidence:** 2

**Summary:**

This paper tackles the task of time series forecasting using masked modeling. The authors propose a patch reconstruction task to facilitate training. The patches are treated independently in the patch reconstruction task. A complimentary contrastive learning, where two views with a 50% masking ratio are used for CL, is utilized. Experiments on the common forecasting benchmark with 7 datasets, and the classification benchmark with 5 datasets, show the efficacy of the proposed method.

**Strengths:**

1. The paper is generally well-written and easy to follow. The method seems straightforward to implement.
2. The experiments are thorough. The proposed method is evaluated on two tasks with a total of 12 datasets. Also, the transfer learning setting is explored. The key PI vs. PD task is analyzed through quantitative and qualitative experiments.

**Weaknesses:**

1. The training and inference efficiency analyses are brief or missing. It would be useful to see whether the patch-independent design can also bring benefits to inference time. The model requires contrastive learning and reconstruction loss, which might drastically increase training time compared to other supervised learning methods. Therefore it would be useful to see it compared to supervised learning methods as well.

**Questions:**

1. What is the input horizon (look-back window/length of historic sequence) for the TSF task (Table 3)? This detail is missing.
2. Are the numbers in Table 3 averaged over multiple runs (with different random seeds) or are they single runs only?

---

> ### Author Response · Authors · 2023-11-18
>
> **W1. The training and inference efficiency analyses are brief or missing**
>
> We agree that the information provided in **Table 2 (of old version)** was somewhat brief, so we extend and move it to **Table 13  (of new version)**. To highlight the additional information, **Table 13  (of new version)** now compares the inference time and performance. **PITS shows ~2.3x faster inference time than PatchTST**, thanks to the efficient PI architecture. Also, we **differentiate our method with or without CL**, such that **Table 13  (of new version)** also shows how much the pretraining time increases to achieve better performance when incorporating the naive or hierarchical CL.
>
> Regarding the concern on **"training time compared to other supervised learning methods"**, we compare the training time of PITS and its supervised learning version (used the PI arch but no pretraining with the PI task)  in **Table I.1 (of new version)**. Self-supervised PITS pretrains the model for 100 epochs, trains the linear head for 10 epochs, and fine-tunes the entire model for 20 epochs in sequence, and supervised PITS trains for 100 epochs to achieve good performance. Given that pretraining is already done, fine-tuning the pretrained model is **x3~4 faster** than training from scratch, while providing better performance.
>
> &nbsp;
>
> **Q1. Size of input horizon for the TSF task (Table 3 of old version; 2 of new version)**
>
> We tuned the input horizon size with 3 candidates via validation: $L \in \\{336, 512, 768\\}$, where we added the information about validation in **Section 4.2**.
>
> We realized that the detail on experimental settings is missing, so we added **Appendix B (of new version)** to provide them. We also clarify that we will release the code upon acceptance for reproducibility. However, if you find some hyperparameter or setting is still not clear, please feel free to share it.
>
> &nbsp;
>
> **Q2. The number of runs for Table 3 (of old version; 2 of new version)**
>
> We ran the experiments 3 times with different random seeds, and they gave us consistent results: the standard deviation is mostly in the order of $10^{-4}$ to $10^{-3}$, while we observe the performance gain in the order of $10^{-2}$ to $10^{-1}$. For more details, we provide the standard deviation of the performance of PITS in **Appendix L (of new version)**.

---

### Official Review · Reviewer_ZBCM · 2023-11-01

**Soundness:** 3 good
**Presentation:** 3 good
**Contribution:** 2 fair
**Rating:** 6
**Confidence:** 3

**Summary:**

Authors propose a mix of contrastive learning and masked modelling on time-series data, and a patch-independent approach to reconstruction is shown to outperform patch dependant (vanilla mask modelling) approach. Their PI approach auto encodes unmanned patches. Authors evaluate on forecasting and classification of time-series data.

**Strengths:**

- Extensive experiments comparing to other methods and ablating different components.
- Incremental but shown to improve approach with simple mechanisms.
- Interesting analysis on distribution shift

**Weaknesses:**

- Novelty is weak, their contribution auto encoding is quite established before masked modelling literature and one of the early approaches to representation learning. It is more that exploring this within time-series data which seems to be there contribution. Also mixing of CL and masked modelling has been explored in other methods but not exactly similar to their approach.


- Missing reference to previous method that Combine CL and MAE but in a different context and different method: Gong, Yuan, et al. "Contrastive audio-visual masked autoencoder." arXiv preprint arXiv:2210.07839 (2022).

**Questions:**

Figure 5 is not as clear what it is conveying? What is Fig. 5 Left showing in terms of colours? Fig. 5 Right it is a bit confusing since it is PD-PI and x & y axis represent the slop and amplitude of diff between training and test phases, It is not clear to me what is exactly being changed to have the distribution shift in the time-series data?

---

> ### Author Response · Authors · 2023-11-18
>
> **W1-a. Contribution of our work**
>
> As noted by Reviewer ZBCM, autoencoding is a well-established pretext task for self-supervised representation learning, prior to masked modeling (MM). Our contribution is on the fact that we **revisit autoencoding by adapting the task and architecture (to be PI) for better TS representation learning**, and our extensive experiments show that the proposed method achieves **SOTA performance while being more efficient than Transformers pretrained with MM** [A,B,C]. We believe our findings shed light on the effectiveness of self-supervised learning through simple pretraining tasks and model architectures in various domains, and provide a strong baseline for TS representation learning.
>
> [A] Zerveas, George, et al. "A transformer-based framework for multivariate time series representation learning." SIGKDD (2021)
>
> [B] Nie, Yuqi, et al. "A time series is worth 64 words: Long-term forecasting with transformers." ICLR (2022)
>
> [C] Dong, Jiaxiang, et al. "SimMTM: A Simple Pre-Training Framework for Masked Time-Series Modeling." NeurIPS (2023).
>
>
>
> &nbsp;
>
>
> **W1-b, W2. References for combining CL and MM**
>
> Thanks for the suggestion, we added a brief discussion on recent works on combining CL and MM in various domains [A,B,C,D,E]. We initially omitted them as our main focus is on PI strategies, but we agree that discussion on those works is indeed interesting and helpful to remind audiences of recent progress.
>
> Among them, we note that SimMTM [E], which is a concurrent work to ours, proposed an MM task with a regularizer in its objective function in the form of a contrastive loss (though they did not claim that the regularizer is about contrastive learning but "series-wise similarity learning" and "point-wise aggregation"). This regularizer enforces that the original TS and its masked TS exhibit close representations and are distinctly different from representations of other series. However, it differs from our work in that it performs CL *between TS*, while our proposed CL operates *within a single TS*. Nevertheless, since our primary emphasis is on PI tasks and architectures, we removed the claim in **Section 1** that "our work is the first to integrate MTM and CL."
>
> [A] Jiang, Ziyu, et al. "Layer Grafted Pre-training: Bridging Contrastive Learning And Masked Image Modeling For Label-Efficient Representations." ICLR (2023).
>
> [B] Yi, Kun, et al. "Masked image modeling with denoising contrast." ICLR (2023).
>
> [C] Huang, Zhicheng, et al. "Contrastive masked autoencoders are stronger vision learners." arXiv preprint arXiv:2207.13532 (2022).
>
> [D] Gong, Yuan, et al. "Contrastive audio-visual masked autoencoder." ICLR (2023)
>
> [E] Dong, Jiaxiang, et al. "SimMTM: A Simple Pre-Training Framework for Masked Time-Series Modeling." NeurIPS (2023).
>
>
> &nbsp;
>
> **Q1. About Figure 5**
>
> The purpose of **Figure 5** is to see ***how robust PI task is under distribution shifts than PD task***. For better understanding, we added further details regarding the explanation of **Figure 5** in **Section 5**.
>
> **[Left Panel]**
>
> The **left panel of Figure 5** depicts a toy dataset comprising 98 time series exhibiting varying degrees of distribution shift, where **we assign a different color to each time series**. The degree of shift is characterized by changes in slope (trend) and amplitude (seasonality).
> About how the toy dataset is exactly made, if the slope and amplitude of the training data are denoted as $s$ and $a$, respectively, then the slope and amplitude of the test data are represented as $r_1 \times s$ and $r_2 \times a$, respectively. Consequently, $r_1$ and $r_2$ serve as indicators of the slope and amplitude ratios, reflecting the degree of distribution shift.
>
> **[Right Panel]**
>
> The **right panel of Figure 5** illustrates the difference in MSE between the model trained with PI and PD tasks using the 98 time series depicted in the left panel (i.e., MSE of PD task - MSE of PI task). The x-axis and y-axis correspond to $r_1$ and $r_2$, respectively. For each $r_1$ and $r_2$, the color intensity shows how much PD is worse than PI (the darker, the worse.)
>
> The observations from this figure are:
> - 1. As the MSE difference (PD-PI) is non-negative for all cases, PI is more robust to distribution shifts than PD. Basically, PD becomes worse than PI as the change of $r_1$ and $r_2$ is larger (out of the center marked with the black rectangle).
> - 2. The color gradient shows in which direction PD becomes even worse: when the slope is flipped (right to left) and the amplitude is increased (top to bottom).
>
> If you have further questions, please feel free to share them. We are happy to address them and improve the clarity of our work.

---

### Author Response · Authors · 2023-11-19

# General Comment

First of all, we deeply appreciate your time and effort in reviewing our paper.
Our work introduces PITS (Patch Independence for Time Series), a self-supervised learning approach for time series representation learning. PITS utilizes patch-independent (PI) strategies in two aspects: (1) **pretraining task** is the simple reconstruction task rather than predicting masked patches from unmasked ones, and (2) **architecture** is the simple MLP rather than Transformer.

&nbsp;

As highlighted by the reviewers, our work is easy to follow and understand (DsGC,aEzH,oFTu), proposing a simple and straightforward method (ZBCM,DsGC) that is novel (aEzH,oFTu) and supported by exhaustive experiments (all). In our responses, we addressed the concerns raised by all reviewers and supplemented our claims with additional analyses. Here we provide highlights to aid your post-rebuttal discussion (all sections/figures/tables labeled based on the revision):

&nbsp;

> **1. More references (DsGC,oFTu)**

DsGC and oFTu expressed concerns about missing references related to methods combining contrastive learning (CL) and masked modeling (MM) (DsGC) and masking strategies in MM (oFTu), respectively. We initially omitted them, but we included relevant references in **Section 2** of the revision as it would be interesting and helpful to the audiences.

&nbsp;

> **2. Comparison with other MM methods in TSF task (aEzH)**

aEzH expressed doubt about the absence of a comparison with other baseline methods for TS forecasting. While additional baseline methods are available, we chose to include only SOTA methods that demonstrated competitive performance and omitted others. In **Section C**, we compare all omitted methods for the TSF task.

&nbsp;

> **3. Efficiency Analysis (DsGC)**

Our initial efficiency analysis in **Table 2 (of old version)** focused solely on efficiency in terms of training time and the number of parameters. Upon request by DsGC, we extended the table to include the inference time and the MSE performance for better comparison. The results in  **Table 13** indicate that our method requires significantly fewer parameters while its training and inference speed is fast compared to PD (PatchTST). Additionally, we compare the efficiency of our method under self-supervised and supervised settings in **Table I.1**.

&nbsp;

> **4. Minor Issues**

Experimental Settings (DsGC, aEzH)
- DsGC and aEzH asked about the details of experimental settings, including hyperparameters. In **Section 4.2** and **Section B**, we provide more details on our experiments.

Detailed explanation on figures/tables (ZBCM, aEzH):
- We have added more explanations for **Figure 2** and **Figure 5**, as noted by aEzH and ZBCM, respectively.

Explanation of limitations and potential challenges (aEzH)
- Though it was not the case in TS benchmark datasets we experimented, there might be some cases where more sophisticated PD strategies outperforms PI strategies. In any case, we hope our proposed PI strategies provide a strong baseline for future TS representation learning works.

Standard deviation of Table 2 (DsGC):
- In **Table 2**, we omitted the standard deviation of the results due to space constraints. However, we included the standard deviation in **Table L.1 (new version)**, where the standard deviation is generally low.

&nbsp;

For your convenience, we uploaded the pdf file containing both the revised manuscript and appendices. Also, in our revised manuscript, we **highlighted the major changes in green.**

Again, we appreciate the discussion and insights provided by the reviewers. We believe they will further strengthen the contribution of our paper, and we will incorporate these insights into the final version. **If we missed something or if you have further questions or suggestions, please share them with us**; we are happy to address them and improve our work.

&nbsp;

Thank you very much.

Authors.

---

### Meta-Review · Area_Chair_ndnW · 2023-12-05

**Metareview:**

The manuscript presents a mix of contrastive learning and masked modelling on time-series data. In doing so, they propose a patch independent approach and show that this PI outperforns patch dependent approach. Extensive experiments on time-series based classification and regression tasks demonstrate the strength of the proposed approach.

The work is interesting and addresses an important problem (Masked modeling and contrastive learning for time-series), and authors also explore the usability in a data drift scenario.

However, the novelty is incremental. Responses to some questions by reviewers are a bit vague. For eg., to reviewer aEzH comment on why the authors think PI is better than PD, the author's responses are rather broad and intutive, instead of being objective.

**Justification For Why Not Higher Score:**

The manuscript needs to be refurbished for better clarity, taking into consideration reviewer's sincere feedbacks. Although this is a good contribution, the novelty of the manuscript is limited.

**Justification For Why Not Lower Score:**

The manuscript addresses an important problem and has a detailed set of empirical evaluations

---

### Decision · Program_Chairs · 2024-01-16

Accept (poster)